# Towards Comprehensive Scene Understanding: Integrating First and Third-Person Views for LVLMs

**Insu Lee**[1][*], **Wooje Park**[1][*], **Jaeyun Jang**[1], **Minyoung Noh**[1],
**Kyuhong Shim**[2][†], **Byonghyo Shim**[1][†]

[1]Seoul National University, [2]Sungkyunkwan University
{islee, wjpark, jyjang, mynoh, bshim}@islab.snu.ac.kr, khshim@skku.edu

## Abstract

Large vision-language models (LVLMs) are increasingly deployed in interactive applications such as virtual and augmented reality, where a first-person (egocentric) view captured by head-mounted cameras serves as key input. While this view offers fine-grained cues about user attention and hand-object interactions, its narrow field of view and lack of global context often lead to failures on spatially or contextually demanding queries. To address this, we introduce a framework that augments egocentric inputs with third-person (exocentric) views, providing complementary information such as global scene layout and object visibility to LVLMs. We present E3VQA, the first benchmark for multi-view question answering with 4K high-quality question-answer pairs grounded in synchronized ego-exo image pairs. Additionally, we propose M3CoT, a training-free prompting technique that constructs a unified scene representation by integrating scene graphs from three complementary perspectives. M3CoT enables LVLMs to reason more effectively across views, yielding consistent performance gains (4.84% for GPT-4o and 5.94% for Gemini 2.0 Flash) over a recent CoT baseline. Our extensive evaluation reveals key strengths and limitations of LVLMs in multi-view reasoning and highlights the value of leveraging both egocentric and exocentric inputs. The dataset and source code are available at `https://github.com/Leeinsu1/Towards-Comprehensive-Scene-Understanding`.

## 1 Introduction

In recent years, large vision-language models (LVLMs) have received significant attention for their unprecedented performance in diverse tasks, including information retrieval, content generation, and multi-modal interaction [47, 18, 17, 13, 25, 16]. Their applications are now extended to interactive and immersive systems like virtual and augmented reality and embodied robotics [5, 24, 28]. Key characteristic of these applications is that images associated with the first-person view (a.k.a. egocentric view), captured by head-mounted cameras or smart glasses, are used as input. Although the first-person view is crucial in revealing the user's intention, LVLMs might not interpret it well since most of the training datasets, often obtained from generic web images, are not acquired from a first-person perspective.

Recently, several approaches overcoming the scarcity of this so called egocentric data have been proposed. Notable ones include synthetic generation of egocentric data, incorporating large-scale egocentric datasets during the pre-training stage, and leveraging parameter-efficient fine-tuning on small-scale egocentric data [23, 33, 36]. While these approaches are effective to some extent, due to inherent constraints of the egocentric view, such as a limited field of view and a lack of global

---

[*] Co-first authors. [†] Co-corresponding authors.

39th Conference on Neural Information Processing Systems (NeurIPS 2025).

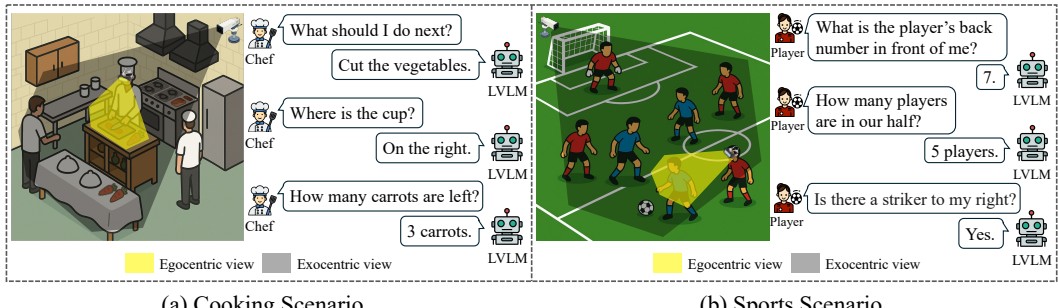

|  (a) Cooking Scenario  |  (b) Sports Scenario  |

Figure 1: Conceptual illustration of example scenarios that require a joint understanding of egocentric (first-person) and exocentric (third-person) views. In each scenario, the first question can be answered using only the egocentric view, while the subsequent two questions require integrating information from both views. Yellow and gray overlays indicate egocentric and exocentric views, respectively.

context, LVLMs might not properly handle user queries requiring broader contextual understanding. Consider the scenario in Figure 1(a), where the user interacts with a visual assistant to make a food. When an egocentric image is provided as an input, LVLMs can readily answer the question "What should I do next?", but may not provide the proper answer to questions such as "Where is the cup?" or "How many carrots are left?", since the egocentric view has limited spatial coverage and thus fails to capture the comprehensive picture of the user environment. The moral of the story is that the egocentric view alone is not sufficient to handle the full range of user queries.

An aim of this paper is to propose a framework integrating multi-view information to enhance LVLMs' capacity to understand the comprehensive context of a user's surrounding environment. The core of our approach is to integrate images from the egocentric view and third-person view (a.k.a. exocentric view) to extract the merits of both. The exocentric view provides broader contextual cues, such as the user's posture and the visibility of objects in blind spots, while the egocentric view offers information about user focus, including hand-object interactions and fine-grained object details. While integrating multiple viewpoints can help answer diverse user queries, it remains unclear whether conventional LVLMs can fully leverage them and really enhance the output quality. To answer the question systematically, we design 1) an ego-exo[1] multi-view question answering benchmark, referred to as the Ego-Exo Expanded Visual Question Answering (E3VQA) benchmark, and 2) a novel prompting technique called Multi-view, Multi-perspective, Multi-turn Chain-of-Thought prompting (M3CoT).

The purpose of E3VQA is to evaluate the ability of LVLMs to jointly perceive and reason when egocentric and exocentric views are provided. To this end, we meticulously curate a dataset comprising 4K question-answer pairs, each coupled with synchronized ego-exo images collected from the Ego-Exo4D dataset [12] covering four distinct tasks, including 1) action understanding, 2) object recognition, 3) spatial reasoning, and 4) numerical estimation. To fully leverage the complementary cues of ego-exo views, we further propose a novel prompting technique called M3CoT. Specifically, we construct a unified description of the scene (i.e., scene graphs) that guides the LVLM to understand the entire scene environment. This process consists of two main stages. First, we build three distinct scene graphs, each capturing the scene from a different perspective. The first graph is generated by simultaneously processing both ego and exo images, providing a holistic view of the scene while potentially overlooking fine-grained details. The second and third graphs are constructed by sequentially processing, either from ego to exo or from exo to ego, allowing the model to capture minute details. Second, we iteratively unify the three scene graphs, enabling each to progressively converge into a more robust and comprehensive representation. These unified scene graphs are used by LVLMs to generate the final responses.

We evaluate M3CoT on the E3VQA benchmark using state-of-the-art LVLMs, including GPT-4o [13] and Gemini 2.0 Flash [37], and observe a considerable gain in accuracy of 4.84% and 5.94% over the recent CoT method CCoT [32]. We also observe the noticeable gain (6.88% and 8.13%) on numerical reasoning questions, demonstrating our method's effectiveness at integrating dual-view information.

In summary, our contributions are as follows:

[1]Ego and exo denote the egocentric and exocentric, respectively.

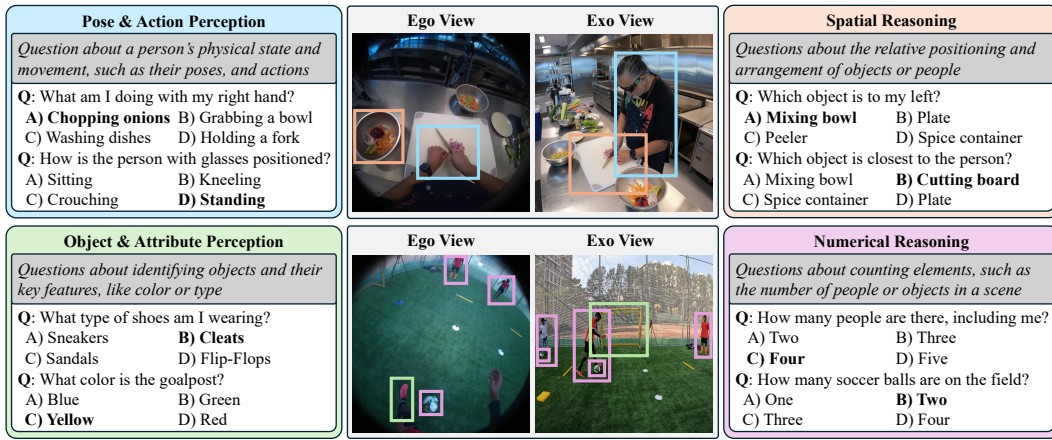

Figure 2: Categories in the E3VQA benchmark. Each question is paired with ego-exo images and multiple-choice answers. The answers are highlighted in bold. The left part shows recognition categories, assessing the ability to focus on question-relevant parts. The right part shows reasoning categories, evaluating the ability to integrate information across views.

- We build the ego-exo multi-view VQA benchmark, E3VQA, consisting of 4K rigorously curated question-answer pairs with synchronized ego-exo image pairs. We construct E3VQA through a systemically designed pipeline to ensure that each instance evaluates the capabilities of LVLMs in integrating and reasoning across ego and exo views.
- To address the challenges posed by E3VQA, we propose M3CoT, a training-free prompting technique that combines scene graphs from three complementary perspectives into a unified scene graph. M3CoT improves the question answering accuracy on the E3VQA benchmark, achieving gains of 4.84% and 5.94% on two leading LVLMs, GPT-4o and Gemini 2.0 Flash.
- We perform a detailed analysis of leading LVLMs on E3VQA, uncovering their specific failure modes in multi-view reasoning and quantifying how egocentric and exocentric inputs affect performance.

## 2 E3VQA Benchmark

### 2.1 Motivation and Objectives

When compared to a single-image setting, LVLMs incorporating multi-view images face a number of challenges. First, the model needs to identify which image and what regions in an image are relevant to the question. Second, the model needs to filter out redundant content appearing in both views. Third, the model should deliberately extract 'complementary' cues from both views to generate a complete answer. Because a single image is provided per question for the conventional visual question answering (VQA) benchmark, they cannot evaluate multi-image reasoning capabilities.

To address this gap, we introduce E3VQA, a multiple-choice benchmark specifically designed for paired ego-exo images. Each question is accompanied by a set of plausible but incorrect options (i.e., *distractors*) that target typical failure patterns. These patterns include relying solely on one image (ego or exo), ignoring visual input altogether, or failing to merge complementary information from both views. These carefully crafted distractors enable E3VQA to precisely evaluate a model's ability to reason across ego-exo image pairs.

### 2.2 E3VQA Composition

We organize E3VQA into four categories: pose and action perception, object and attribute perception, numerical reasoning, and spatial reasoning, to encompass a wide range of real-world scenarios (see Figure 2). Each category contains 1,000 question-answer (QA) pairs, evenly divided between egocentric and exocentric questions (e.g., "What am I doing?" vs. "What is the person doing?"), which supports the evaluation of the model's generalization capability to diverse forms of user queries.

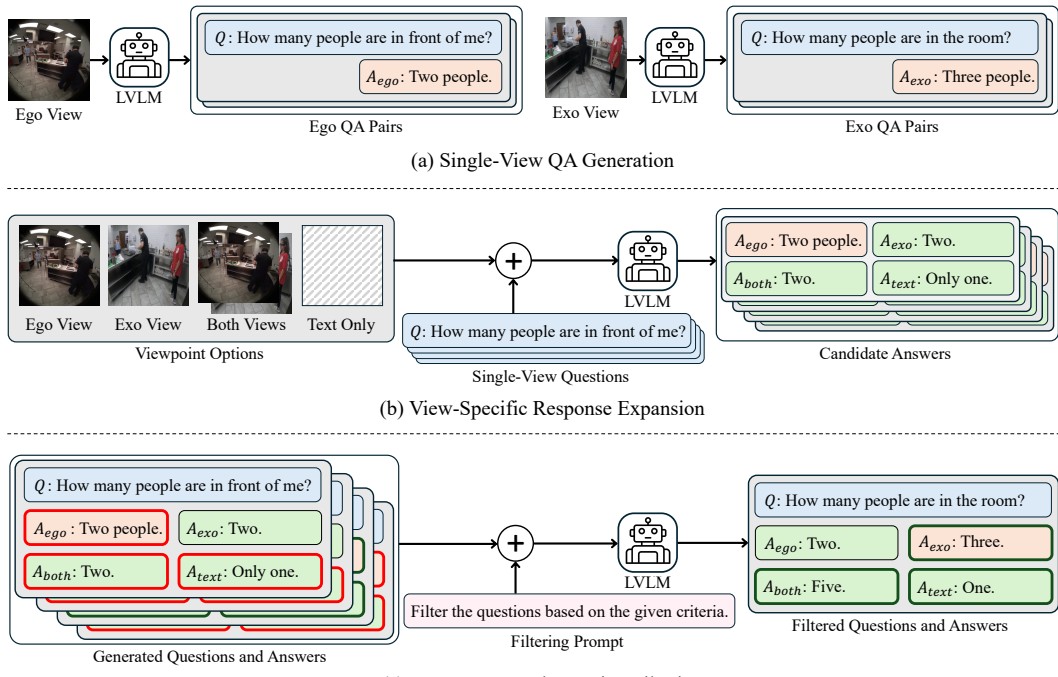

**(a) Single-View QA Generation**

**(b) View-Specific Response Expansion**

**(c) Response-Based Question Filtering**

Figure 3: Overview of the E3VQA benchmark's three-step automated QA generation pipeline: (a) single-view QA generation step, (b) view-specific response expansion step, and (c) response-based question filtering step.

To solve the questions, the model must identify the relevant object or person in both views and align their features across the two images. Variations in viewpoints and fields of view can make the same element look distorted, partially occluded, or differently scaled, making it difficult to establish correspondences and integrate visual cues. Detailed explanations of the challenges for each category and dataset statistics are provided in Appendices A.1 and A.2.

## 2.3 Dataset Construction Pipeline

### 2.3.1 Source Data Pre-Processing

The E3VQA benchmark is constructed utilizing the large-scale synchronized ego-exo dataset, Ego-Exo4D [12], which is composed of diverse user interactions (e.g., cooking, bike repair, soccer) filmed in various environments and countries. To ensure diversity within the dataset, video clips are uniformly sampled from the Ego-Exo4D dataset with respect to both user activities and recording locations. Each selected video clip is downsampled to 1 frame per second, and 8 frames are uniformly sampled from the downsampled clip. As a result, 4,600 ego-exo image pairs are obtained from the 575 video clips. Note that all video clips are selected from the test split to prevent any potential dataset contamination.

### 2.3.2 Automated QA Generation

We introduce a three-step automated QA generation pipeline designed to minimize human effort and improve scalability (illustrated in Figure 3). Throughout the entire process, we utilize GPT-4o [13], a powerful off-the-shelf LVLM.

**Step 1: Single-View QA Generation** We begin by generating QA pairs independently from either the ego or exo images, under the assumption that recent LVLMs show a stronger understanding of single images than of multi-view inputs. Specifically, we instruct the model to generate Ego QA pairs (e.g., $Q$: What am I doing?, $A_{ego}$: Pouring water) from the ego images, and Exo QA pairs (e.g., $Q$:

What is the person doing?, $A_{exo}$ : Stirring eggs) from the exo images. This design ensures that the generated questions align with the visual characteristics of each view: the ego image enables clear identification of the 'I', while the exo image provides a broader field of view and richer contextual information about 'the person'. For each image, we generate three QA pairs per category to balance diversity and relevance; generating more than three often results in questions that are not visually grounded. As a result, this process yields a total of 110,400 single-view QA pairs.

**Step 2: View-Specific Response Expansion**  In this step, we generate diverse candidate answers by presenting the model with the same question under four distinct input conditions: 1) ego view only, 2) exo view only, 3) both ego and exo views, and 4) text only (no visual input). The resulting answers are denoted by $A_{ego}$, $A_{exo}$, $A_{both}$, and $A_{text}$, respectively. These responses serve two key purposes in subsequent steps: 1) they are used as criteria for identifying low-quality QAs during the filtering stage, and 2) they function as hard candidate options for constructing multiple-choice questions in the human verification stage.

**Step 3: Response-Based Question Filtering**  A large proportion of the questions generated in the previous step are either too easy or disqualified for multi-view question answering. To filter out such questions at scale, we introduce a response-based question filtering strategy that uses the initial answer from Step 1 (denoted as $A_{init}$; either $A_{ego}$ or $A_{exo}$) as a reference. First, we discard questions where $A_{text}$ matches $A_{init}$, since this indicates that the question can be answered without any visual input. Second, we remove questions for which $A_{both}$ matches $A_{init}$, which indicates that the answer remains unchanged even when both ego and exo views are provided, and therefore implies that multi-view reasoning is unnecessary. Applying these two criteria retains only those questions that cannot be answered without integrating both views. As a result, approximately 78.5% of the initial questions are filtered out, leaving a set of 23,694 challenging, multi-view QA samples.

### 2.3.3   Human Verification

Following automated QA generation, we perform a human verification with four expert annotators. The experts thoroughly review all questions, discarding unclear or low-quality questions. They also carefully craft the answer options, leveraging the responses generated in the previous step: $A_{ego}$, $A_{exo}$, $A_{both}$, and $A_{text}$. This process results in the final E3VQA dataset, comprising 4,000 high-quality QA pairs, representing just 3.6% of the original 110,400 samples after filtering and refinement. Please refer to Appendix A.3 for additional details.

## 3   M3CoT: Multi-Perspective Scene Understanding

### 3.1   Multi-Perspective Scene Graph Generation

In our proposed ego-exo multi-image question answering scenario, we expect the LVLM to generate the most appropriate answer given a query $Q$ and a pair of ego and exo images $\mathbf{I} = \{I_{\text{ego}}, I_{\text{exo}}\}$. To help the model understand ego and exo images comprehensively, we employ a multi-perspective scene graph generation approach. Specifically, three instances of an LVLM act as distinct agents, denoted as $\mathcal{F}_1$, $\mathcal{F}_2$, and $\mathcal{F}_3$, each generating a scene graph from a different perspective as follows:

- **Ego&Exo**: Agent $\mathcal{F}_1$ generates a joint scene graph $S_1$ in a single step by simultaneously processing both $I_{\text{ego}}$ and $I_{\text{exo}}$ as input.
- **Ego2Exo**: Agent $\mathcal{F}_2$ first generates a scene graph using only $I_{\text{ego}}$, which is then sequentially expanded by incorporating information from $I_{\text{exo}}$ to generate scene graph $S_2$.
- **Exo2Ego**: Agent $\mathcal{F}_3$ follows the reverse approach, generating a scene graph based solely on $I_{\text{exo}}$ and subsequently supplementing it with $I_{\text{ego}}$ to generate scene graph $S_3$.

Together, the three agents can capture both view-specific details and holistic scene context from complementary perspectives. For each perspective, prompts are carefully designed to capture the complementary information present in the ego-exo images. Please refer to Appendix E.2 for the complete prompts.

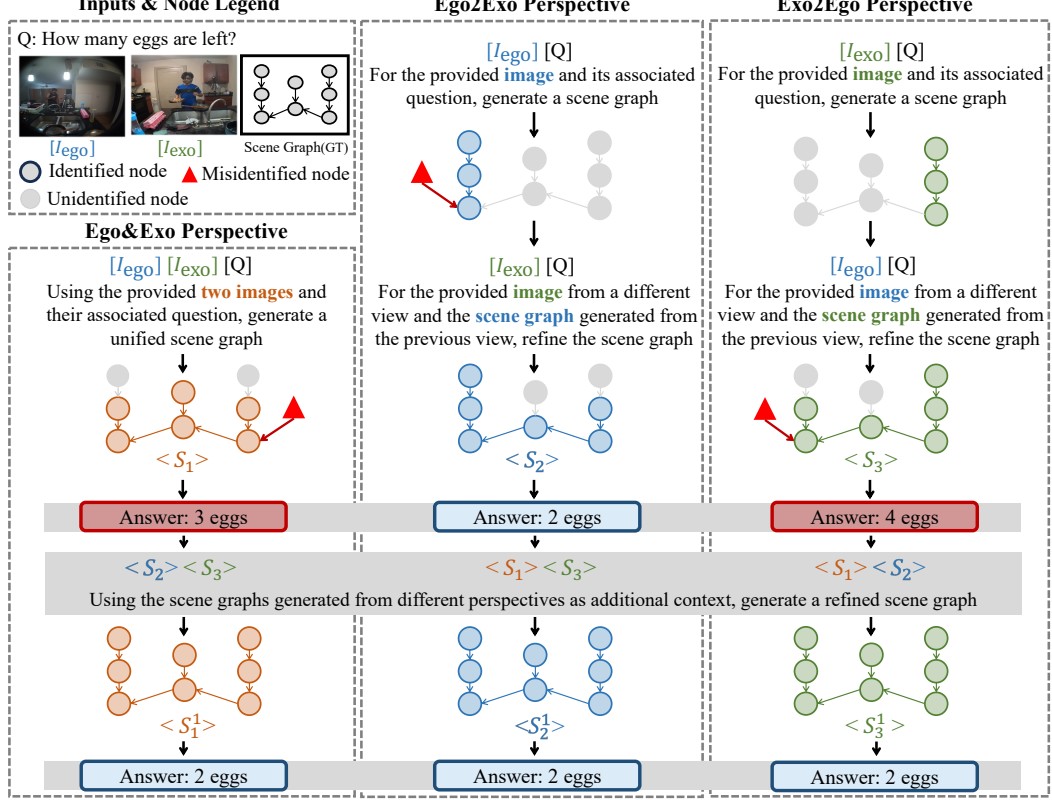

Figure 4: Overview of the M3CoT method. **Left:** Scene graph generation process from the Ego&Exo perspectives. **Center:** Scene graph generation process from the Ego2Exo perspective. **Right:** Scene graph generation process from the Exo2Ego perspective. Scene graphs from each perspective are merged to complement missing objects and relations, enabling the model to perform coherent reasoning and answer generation.

## 3.2 Iterative Multi-Agent Scene Graph Refinement

To further refine each scene graph $S_i$ (for $i = 1, 2, 3$) generated by agent $\mathcal{F}_i$, we iteratively incorporate information from the other two agents. At iteration $t$, agent $\mathcal{F}_i$ takes the other two scene graphs (e.g., $S_2^{t-1}$ and $S_3^{t-1}$ for $\mathcal{F}_1$) to examine their objects, attributes, and relationships, and then adjusts $S_i^{t-1}$ to $S_i^t$ to better align with both $I_{\text{ego}}$ and $I_{\text{exo}}$. Here, $S_i^t$ denotes the scene graph $S_i$ after the $t$-th update. By leveraging complementary information from multiple perspectives, this process improves both the accuracy and completeness of each agent's scene graph. At each iteration step $t$, every agent $\mathcal{F}_i$ generates an answer to the question, conditioned on $S_i^t$. We then aggregate the agents' responses via majority voting. If a consensus is achieved, we accept the majority answer and terminate the process. If the agents' answers remain inconsistent after a fixed number of iterations, the final answer is selected from the response of $\mathcal{F}_1$. This iterative loop yields progressively richer scene representations and promotes convergence among the agents' answers.

## 4 Experimental Results

### 4.1 LVLMs' Performance on E3VQA

To assess ego-exo multi-image reasoning capabilities, we evaluate five closed-source and nine open-source LVLMs on the E3VQA benchmark using their default configurations. Detailed model specifications, experimental settings, as well as system and user prompt templates are provided in Appendices B.1 and E.2.

Table 1: Performance comparison of recent closed and open-source models on the E3VQA benchmark.

| LVLMs | Pose & Action | | Object & Attribute | | Numerical | | Spatial | | Avg. |
|---|---|---|---|---|---|---|---|---|---|
| | Ego | Exo | Ego | Exo | Ego | Exo | Ego | Exo | |
| **Closed-Source** | | | | | | | | | |
| GPT-4o [13] | $49.87_{\pm 1.10}$ | $\mathbf{63.47}_{\pm 0.81}$ | $72.47_{\pm 0.12}$ | $\mathbf{77.00}_{\pm 0.40}$ | $\mathbf{48.47}_{\pm 1.14}$ | $57.67_{\pm 0.12}$ | $\mathbf{61.13}_{\pm 0.99}$ | $57.13_{\pm 0.12}$ | $\mathbf{60.90}$ |
| GPT-4o mini [13] | $41.33_{\pm 0.31}$ | $49.00_{\pm 0.40}$ | $66.07_{\pm 0.12}$ | $71.00_{\pm 0.00}$ | $35.80_{\pm 0.35}$ | $44.00_{\pm 0.00}$ | $44.00_{\pm 0.20}$ | $41.00_{\pm 0.20}$ | $49.03$ |
| Gemini 2.0 Flash [37] | $53.27_{\pm 0.12}$ | $60.33_{\pm 0.12}$ | $\mathbf{74.00}_{\pm 0.20}$ | $76.47_{\pm 0.50}$ | $46.33_{\pm 0.42}$ | $56.20_{\pm 0.20}$ | $58.27_{\pm 0.46}$ | $53.80_{\pm 0.20}$ | $59.80$ |
| Gemini 1.5 Pro [11] | $\mathbf{53.73}_{\pm 0.46}$ | $62.40_{\pm 1.31}$ | $69.60_{\pm 1.31}$ | $72.27_{\pm 0.83}$ | $44.07_{\pm 1.01}$ | $52.60_{\pm 1.00}$ | $58.27_{\pm 2.14}$ | $54.00_{\pm 1.00}$ | $58.37$ |
| Claude 3.5 Sonnet [1] | $40.33_{\pm 0.42}$ | $50.6_{\pm 0.20}$ | $59.13_{\pm 0.90}$ | $62.00_{\pm 0.53}$ | $44.13_{\pm 1.03}$ | $49.4_{\pm 1.25}$ | $50.73_{\pm 3.19}$ | $41.20_{\pm 0.60}$ | $49.69$ |
| **Open-Source** | | | | | | | | | |
| InternVL3-14B[55] | $44.73_{\pm 1.50}$ | $54.93_{\pm 1.42}$ | $68.13_{\pm 0.81}$ | $73.73_{\pm 0.99}$ | $35.60_{\pm 1.11}$ | $53.00_{\pm 0.20}$ | $45.67_{\pm 0.58}$ | $\mathbf{48.33}_{\pm 0.99}$ | $\mathbf{53.02}$ |
| Qwen2.5-VL-7B [3] | $50.87_{\pm 0.23}$ | $53.33_{\pm 0.23}$ | $\mathbf{69.60}_{\pm 0.20}$ | $\mathbf{75.93}_{\pm 0.46}$ | $\mathbf{35.93}_{\pm 0.12}$ | $47.87_{\pm 0.31}$ | $46.07_{\pm 0.12}$ | $41.27_{\pm 0.23}$ | $52.61$ |
| Qwen2-VL-7B [39] | $\mathbf{53.67}_{\pm 0.90}$ | $\mathbf{56.07}_{\pm 1.72}$ | $67.13_{\pm 0.76}$ | $67.47_{\pm 1.14}$ | $32.87_{\pm 0.99}$ | $38.07_{\pm 2.42}$ | $43.27_{\pm 1.72}$ | $42.27_{\pm 1.01}$ | $50.10$ |
| LLaVA-OneVision-7B [20] | $39.87_{\pm 0.12}$ | $50.73_{\pm 0.64}$ | $67.60_{\pm 0.69}$ | $68.80_{\pm 0.35}$ | $34.87_{\pm 0.64}$ | $40.87_{\pm 0.31}$ | $\mathbf{49.20}_{\pm 0.20}$ | $42.93_{\pm 0.70}$ | $49.36$ |
| InternVL2-8B[38] | $42.20_{\pm 0.53}$ | $44.20_{\pm 0.53}$ | $61.67_{\pm 0.31}$ | $64.67_{\pm 0.23}$ | $33.40_{\pm 0.53}$ | $38.53_{\pm 0.76}$ | $43.67_{\pm 0.50}$ | $41.13_{\pm 0.31}$ | $46.18$ |
| LLaVA-NeXT-7B [21] | $34.67_{\pm 0.92}$ | $33.87_{\pm 0.46}$ | $57.33_{\pm 0.12}$ | $62.27_{\pm 0.23}$ | $30.27_{\pm 0.23}$ | $39.07_{\pm 0.46}$ | $47.20_{\pm 0.35}$ | $40.67_{\pm 0.58}$ | $43.17$ |
| Mantis-8B-Idefics2 [15] | $28.07_{\pm 0.70}$ | $35.47_{\pm 0.23}$ | $53.73_{\pm 0.12}$ | $56.53_{\pm 0.42}$ | $35.67_{\pm 0.31}$ | $37.73_{\pm 0.64}$ | $41.53_{\pm 0.23}$ | $32.53_{\pm 0.90}$ | $40.16$ |
| Deepseek-VL-Chat-7B [27] | $32.60_{\pm 0.72}$ | $34.27_{\pm 0.42}$ | $51.80_{\pm 0.53}$ | $52.47_{\pm 0.23}$ | $32.80_{\pm 0.87}$ | $29.80_{\pm 0.53}$ | $41.00_{\pm 0.40}$ | $36.60_{\pm 1.22}$ | $38.92$ |
| Qwen-VL-Chat-7B [2] | $25.20_{\pm 1.04}$ | $26.60_{\pm 0.92}$ | $33.60_{\pm 1.11}$ | $36.80_{\pm 1.22}$ | $21.07_{\pm 2.69}$ | $21.73_{\pm 1.03}$ | $29.47_{\pm 1.70}$ | $30.53_{\pm 0.76}$ | $28.13$ |

Table 1 reports model accuracy across categories, each consisting of 500 egocentric (Ego) and 500 exocentric (Exo) questions. Even the best-performing model, GPT-4o, achieves only 60.90% accuracy on E3VQA, underscoring the benchmark's difficulty. Among open-source models, InternVL3-14B attains the highest accuracy, while Qwen2.5-VL-7B delivers competitive performance despite its smaller number of parameters. Overall, LVLMs struggle the most with numerical reasoning yet perform relatively well on object and attribute recognition. Notably, models consistently underperform on egocentric questions compared to exocentric questions, highlighting difficulties in resolving the first-person perspective.

## 4.2 Performance Evaluation of M3CoT

To demonstrate the effectiveness of our M3CoT prompting scheme, we compare our technique with three recent multimodal CoT techniques (DDCoT, CoCoT, and CCoT) on the E3VQA benchmark. Experiments are conducted using two leading LVLMs, GPT-4o and Gemini 2.0 Flash, both of which achieve the best performance on E3VQA. Table 2 presents category-wise accuracy for egocentric and exocentric questions. M3CoT improves over CCoT by 4.84 % on GPT-4o and 5.94 % on Gemini 2.0 Flash. In addition, it surpasses DDCoT and CoCoT by 4.15% and 5.71% on GPT-4o, and by 5.03% and 5.81% on Gemini, respectively. The substantial gains in Numerical Reasoning (6.88% and 8.13% over CCoT) highlight M3CoT's ability to integrate multi-view information for a more complete and accurate understanding. In contrast to existing methods, which exhibit limited or inconsistent improvements and occasionally even show performance drops, M3CoT achieves consistent and substantial gains across all categories. These results validate the effectiveness of our approach in addressing the limitations of current multimodal CoT techniques. For additional results on open-source LVLMs, please refer to Appendix C.2.

## 5 Analysis

### 5.1 Analysis of Automated QA Generation Pipeline

To examine how the source of distractors affects the question difficulty, we sample 160 questions (40 per category) and construct four alternative option sets. In each set, all four answer choices are drawn from a single source: text-only, ego view, exo view, or both views. This setup contrasts with our standard configuration, where each distractor is drawn from a different source. As shown in Figure 5(a), the model's error rate increases in the following order: text-only, both-view, single-view, and our composite setting. This result highlights that constructing answer options from diverse

Table 2: Performance comparison of recent multimodal CoT methods on top-performing models.

| Methods | Pose & Action | | Object & Attribute | | Numerical | | Spatial | | Avg. |
|---|---|---|---|---|---|---|---|---|---|
| | Ego | Exo | Ego | Exo | Ego | Exo | Ego | Exo | |
| **GPT-4o** | | | | | | | | | |
| Default | $49.87_{\pm 1.10}$ | $63.47_{\pm 0.81}$ | $72.47_{\pm 0.12}$ | $77.00_{\pm 0.40}$ | $48.47_{\pm 1.14}$ | $57.67_{\pm 0.12}$ | $61.13_{\pm 0.99}$ | $57.13_{\pm 0.12}$ | 60.90 |
| DDCoT [54] | $55.20_{\pm 0.72}$ | $\mathbf{69.53}_{\pm 0.31}$ | $73.80_{\pm 0.92}$ | $78.80_{\pm 0.40}$ | $48.13_{\pm 0.64}$ | $57.87_{\pm 0.58}$ | $67.27_{\pm 0.76}$ | $\mathbf{64.87}_{\pm 0.50}$ | 64.43 |
| CoCoT [50] | $50.93_{\pm 0.31}$ | $66.80_{\pm 0.69}$ | $72.20_{\pm 0.69}$ | $76.33_{\pm 0.64}$ | $49.93_{\pm 1.75}$ | $60.93_{\pm 0.90}$ | $65.07_{\pm 1.10}$ | $60.80_{\pm 0.40}$ | 62.87 |
| CCoT [32] | $55.53_{\pm 0.81}$ | $67.47_{\pm 0.31}$ | $73.00_{\pm 0.69}$ | $77.67_{\pm 0.61}$ | $48.27_{\pm 1.86}$ | $62.27_{\pm 0.12}$ | $63.73_{\pm 0.90}$ | $62.00_{\pm 0.53}$ | 63.74 |
| **M3CoT (Ours)** | $\mathbf{58.40}_{\pm 0.28}$ | $69.40_{\pm 1.13}$ | $\mathbf{78.90}_{\pm 0.42}$ | $\mathbf{82.80}_{\pm 1.70}$ | $\mathbf{56.40}_{\pm 1.13}$ | $\mathbf{67.90}_{\pm 0.71}$ | $\mathbf{71.90}_{\pm 2.69}$ | $62.90_{\pm 0.14}$ | **68.58** |
| **Gemini 2.0 Flash** | | | | | | | | | |
| Default | $53.27_{\pm 0.12}$ | $60.33_{\pm 0.12}$ | $74.00_{\pm 0.20}$ | $76.47_{\pm 0.50}$ | $46.33_{\pm 0.42}$ | $56.20_{\pm 0.20}$ | $58.27_{\pm 0.46}$ | $53.80_{\pm 0.20}$ | 59.80 |
| DDCoT [54] | $55.60_{\pm 0.72}$ | $62.60_{\pm 1.04}$ | $75.53_{\pm 0.95}$ | $81.13_{\pm 0.95}$ | $46.13_{\pm 0.95}$ | $54.67_{\pm 1.29}$ | $57.47_{\pm 0.42}$ | $55.60_{\pm 1.22}$ | 61.09 |
| CoCoT [50] | $55.40_{\pm 0.40}$ | $61.67_{\pm 0.50}$ | $73.80_{\pm 1.25}$ | $77.93_{\pm 0.12}$ | $45.27_{\pm 1.21}$ | $56.20_{\pm 0.35}$ | $58.67_{\pm 0.42}$ | $53.53_{\pm 1.01}$ | 60.31 |
| CCoT [32] | $55.93_{\pm 0.46}$ | $61.53_{\pm 0.31}$ | $71.47_{\pm 0.76}$ | $76.93_{\pm 0.58}$ | $46.67_{\pm 0.70}$ | $60.73_{\pm 1.10}$ | $57.27_{\pm 2.27}$ | $50.93_{\pm 0.61}$ | 60.18 |
| **M3CoT (Ours)** | $\mathbf{57.80}_{\pm 0.20}$ | $\mathbf{65.80}_{\pm 0.20}$ | $\mathbf{78.80}_{\pm 0.72}$ | $\mathbf{82.80}_{\pm 0.40}$ | $\mathbf{55.60}_{\pm 1.06}$ | $\mathbf{67.40}_{\pm 1.25}$ | $\mathbf{62.67}_{\pm 0.81}$ | $\mathbf{58.07}_{\pm 1.10}$ | **66.12** |

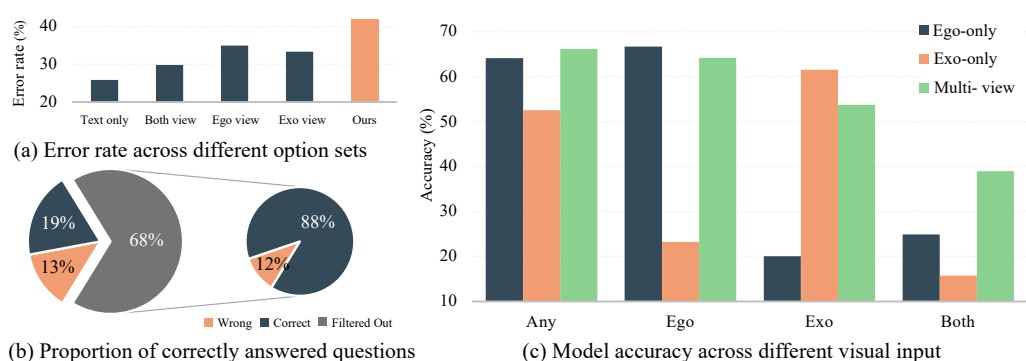

(a) Error rate across different option sets

(b) Proportion of correctly answered questions

(c) Model accuracy across different visual input

Figure 5: Analysis of the benchmark construction pipeline and model performance under varied input conditions: (a) error rate across option-generation strategies, (b) proportion of correctly answered questions between retained and excluded questions, and (c) performance across different visual input modalities.

input sources increases question difficulty, thereby providing a more rigorous evaluation of LVLMs' multi-view reasoning abilities.

To assess the effectiveness of our question filtering process, we compare model accuracy on two subsets: a filtered subset (68% of the data) removed during our filtering process and an unfiltered subset (the remaining 32%). As shown in Figure 5(b), 42% of questions in the unfiltered subset are answered incorrectly with ego-exo multi-view input, whereas only 12% of questions in the filtered subset are answered incorrectly. This substantial gap indicates that the filtering pipeline effectively removes questions solvable by superficial cues while retaining more challenging ones that better evaluate a model's reasoning ability. As a result, the subsequent human verification process becomes more reliable and efficient.

## 5.2 Analysis of LVLM Performance under Single and Multi-View Inputs

We partition the E3VQA benchmark into four subsets, Any, Ego, Exo, and Both, based on which view(s) are required to answer each question. Specifically, questions in the Any subset can be answered using either the egocentric or exocentric image alone, as each view individually contains sufficient information. The Ego and Exo subsets, in turn, require only the egocentric or exocentric image, respectively, since all relevant information appears in a single view. The Both subset requires integrating cues from both egocentric and exocentric views. We then investigate how the model's responses differ between single-view and multi-view inputs within each subset. As shown in Figure 5(c), in the Any subset, providing both images yields a marginal accuracy gain, indicating that

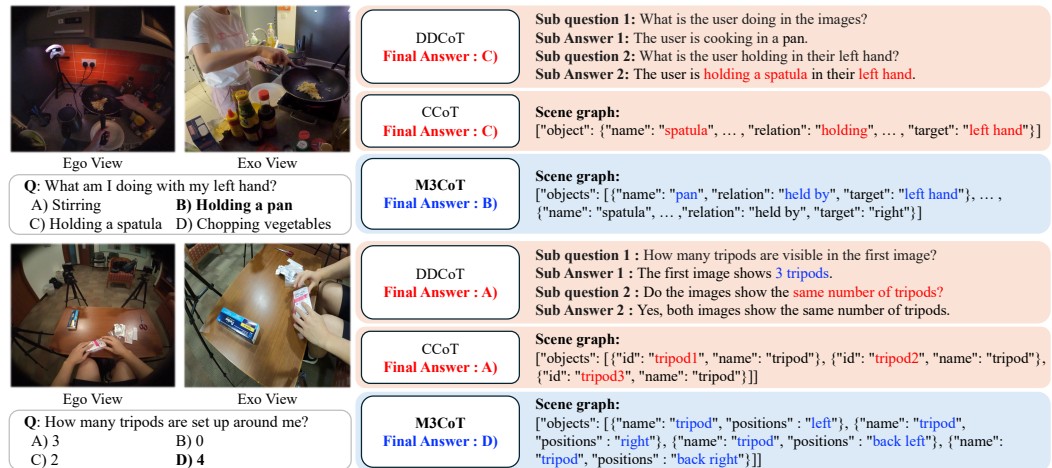

Figure 6: Qualitative examples of answers and reasoning processes generated by different prompting methods. Blue/Red words indicate the key cues that lead to correct/wrong answers.

consistent cues across views can help reinforce the model's prediction of the correct answer. In the Ego and Exo subsets, providing both images leads to a noticeable performance drop, suggesting that redundant context may confuse the model. In the Both subset, multi-view inputs improve accuracy compared to single-view setups; however, performance remains low, staying below 40%.

## 5.3   Analysis of Multi-Perspective Scene Graph Generation Strategies

To analyze the advantages of our three scene graph generation perspectives (Ego&Exo, Ego2Exo, and Exo2Ego), we report in Table 3 their respective performances across the Any, Ego, Exo, and Both subsets. The Ego&Exo strategy achieves the largest accuracy gain in the Both subset, demonstrating its strength in integrating complementary cues across viewpoints. In contrast, Ego2Exo performs best on the Exo subset, while Exo2Ego yields the highest improvement on the Ego subset, reflecting their specialization in inferring one view from the other. These results highlight that different scene graph generation strategies provide complementary advantages depending on where the key information required to answer the question is located within the ego-exo view images. In addition, our scene graph refinement stage improves performance beyond any individual strategy by combining their strengths and compensating for their limitations. Overall, these findings confirm that fusing diverse scene graph perspectives produces more robust reasoning in ego-exo multi-view scenarios.

## 5.4   Qualitative Examples

Figure 6 shows qualitative examples of answers and reasoning processes generated by DDCoT, CCoT, and M3CoT methods using Gemini 2.0 Flash. Although CCoT produces plausible scene graphs, it fails to integrate information across multiple views, resulting in incorrect answers. Specifically, it often misidentifies the same object observed from different perspectives as separate entities. In contrast, our method effectively extracts key information, aligns observations across views, and accurately identifies the same object to answer the question through a multi-perspective, multi-turn reasoning process.

Table 3: Performance comparison of M3CoT's three perspectives across subsets grouped by the image view(s) required to answer.

| Perspective | Required View(s) | | | | Avg. |
|---|---|---|---|---|---|
| | Any | Ego | Exo | Both | |
| Default | 66.29 | 64.83 | 54.21 | 37.49 | 59.80 |
| Ego&Exo | 66.29 | 67.44 | 56.67 | 50.87 | 63.65 |
| Ego2Exo | 68.08 | 62.71 | 61.51 | 43.91 | 62.83 |
| Exo2Ego | 66.94 | 68.02 | 59.92 | 39.13 | 62.98 |
| **M3CoT (ours)** | **69.79** | **69.28** | **62.91** | **53.04** | **66.12** |

# 6 Related Work

## 6.1 Ego-Exo Datasets and Tasks

Egocentric and exocentric views offer complementary information for understanding users and their environments. Early datasets like Charades-Ego [35] and LEMMA [14] introduced paired ego-exo data, while Ego-Exo4D [12] further scaled this paired ego-exo data with large, synchronized videos capturing diverse real-world scenarios. To generalize semantic understanding across multiple perspectives, a body of work has focused on learning view-invariant representations [40, 45]. Furthermore, efforts to align ego-exo content have emerged, including object-level mappings [10] and techniques for identifying and segmenting camera wearers in exocentric scenes [9, 53]. In parallel, cross-view knowledge transfer has been actively explored, with each perspective leveraged to improve the understanding of the other [51, 22, 43, 34]. Several studies have addressed viewpoint selection across perspectives by proposing methods for dynamically selecting informative views over time [29, 30]. Others have explored generating egocentric videos from exocentric video inputs using diffusion-based models [26, 44] or cropping exocentric image frames to distill egocentric-relevant cues [6]. Despite these advances, a task that jointly reasons over synchronized egocentric and exocentric views within LVLMs remains underexplored, highlighting a promising direction for future research.

## 6.2 Visual Question Answering with LVLMs

Visual Question Answering (VQA) benchmarks test LVLMs' ability to interpret and reason over diverse visual content. Most existing VQA benchmarks are constructed from large-scale web-crawled data, typically consisting of images captured from fixed third-person cameras [42, 48, 49]. To support scenarios that require understanding from the user's perspective, egocentric VQA benchmarks capturing first-person views have been introduced. EgoVQA [8] evaluates first-person visual understanding capabilities by offering both egocentric and exocentric queries on first-person visual inputs. EgoSchema [31] evaluates long-form egocentric video understanding by assessing a model's ability to recall previously observed objects and events. EgoThink [4] evaluates first-person reasoning capabilities across diverse categories that reflect practical real-world scenarios. Another line of work includes embodied QA benchmarks such as EmbodiedQA [5] and OpenEQA [28], where agents are required to navigate or interact with their environments to answer given queries. Although numerous VQA benchmarks aim to evaluate LVLMs across diverse aspects, no existing benchmark assesses a model's ability to seamlessly combine complementary visual information from paired ego and exo views. For a detailed comparison of VQA benchmarks, please refer to Table 4 in the Appendix.

## 6.3 Chain-of-Thought Prompting in LVLMs

Building on its success in large language models (LLMs), Chain-of-Thought (CoT) prompting has been extended to LVLMs to enhance inference-time reasoning. DDCoT breaks down a question into a sequence of sub-questions and corresponding sub-answers, which are then used collectively to derive the final answer to the original question [54]. CoCoT, introduced for multi-image input scenarios, compares the similarities and differences between images, guiding the model to answer questions based on the identified visual contrasts [50]. CCoT facilitates understanding the overall context of an image through scene graphs, which are first generated by the LVLM and then incorporated into the prompt to enable compositional reasoning over objects, relations, and attributes [32]. Despite their successes, their applicability to ego-exo multi-image contexts remains unexplored, raising an open challenge for extending CoT methods to these settings.

# 7 Conclusion

In this work, we introduced E3VQA, the first benchmark that systematically assesses whether LVLMs can reason jointly over egocentric and exocentric views. By curating 4K high-quality question-answer pairs grounded in synchronized ego-exo images, E3VQA serves as a rigorous testbed for multi-view understanding. In addition, we proposed M3CoT, a novel prompting strategy that merges scene graphs from diverse perspectives into a unified graph. Extensive experiments demonstrate that M3CoT consistently outperforms the strong CCoT baseline, underscoring the importance of multi-perspective integration for multi-view understanding. By establishing a benchmark for evaluating LVLMs' ego-exo reasoning and enhancing their multi-view understanding ability, this work takes a step toward more context-aware visual assistants capable of operating in complex, real-world environments.

## Acknowledgments

This work was supported by the Institute of Information & communications Technology Planning & Evaluation (IITP) (RS-2024-00398157) and IITP under the artificial intelligence semiconductor support program to nurture the best talents (IITP-2023-RS-2023-00256081), and by the National Research Foundation of Korea (NRF) (RS-2023-00208985), all funded by the Korea government (MSIT).

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

Table 4: Comparison between E3VQA and existing VQA benchmarks. Unlike other benchmarks, E3VQA is designed to evaluate comprehensive scene understanding and reasoning across diverse question perspectives using paired ego-exo images.

| Benchmark | Task Objective | Visual Perspective | Question Perspective | Answer Type | Evaluator | #Questions (test) |
|---|---|---|---|---|---|---|
| MSVD-QA [42] | General Understanding | Exo | Exo | Predefined-Label | Accuracy | 13K |
| MSRVTT-QA [42] | General Understanding | Exo | Exo | Predefined-Label | Accuracy | 72K |
| Social-IQ [49] | Social Understanding | Exo | Exo | Multi-Choice | Accuracy | 7.5K |
| Pano-AVQA [48] | Spatial / Audio-Visual Reasoning | Exo | Exo | Predefined-Label | Accuracy | 5.3K |
| EgoVQA [8] | Egocentric Visual Understanding | Ego | Ego or Exo | Multi-Choice | Accuracy | 120 |
| EgoSchema [31] | Long-Term Reasoning | Ego | Ego | Multi-Choice | Accuracy | 5K |
| EgoThink [4] | First-Person Thinking | Ego | Ego | Open-Ended | LLMs | 700 |
| EmbodiedQA [5] | Goal-Driven Scene Understanding | Ego | Exo | Predefined-Label | Accuracy | 529 |
| OpenEQA [28] | Environment Understanding | Ego | Ego or Exo | Open-Ended | LLMs | 1.6K |
| **E3VQA** | Comprehensive Scene Understanding and Reasoning | Ego and Exo | Ego or Exo | Multi-Choice | Accuracy | 4K |

# A   Additional Details of the E3VQA Benchmark

## A.1   Categories and Challenges

In addition to the challenges described in Section 2.2, each of the following four categories highlights a distinct challenge in the ego-exo multi-image scenario:

- **Pose & Action Perception** focuses on recognizing a person's physical state and movement, such as how their body is positioned and what kinds of gestures or actions they are performing. The presence of multiple people, including the user and duplicated individuals across views, can confuse the model when identifying the question's target. The model must correctly identify the intended individuals and interpret their physical state and behavior.

- **Object & Attribute Perception** involves identifying objects and their attributes, such as color, pattern, or type. Objects may appear in only one view, be partially occluded, or look different due to variations in viewpoint and field of view. To answer correctly, models must resolve such ambiguities and ground the object consistently across views.

- **Numerical Reasoning** addresses tasks involving counting and comparing quantities, such as determining the number of people or objects in a scene. A single view may not include all instances necessary to answer the question, and the same object may appear redundantly across different views. To produce accurate counts, the model must integrate information from both views by handling overlapping objects and aggregating evidence across views.

- **Spatial Reasoning** focuses on understanding the spatial information of a scene, including how objects and people are positioned relative to one another and how they are arranged within the environment. In multi-view spatial reasoning, differences in viewpoint angle and field of view can cause the same object to appear at varying positions in each image, become occluded in some views, or exhibit different spatial relationships with surrounding objects. To overcome these challenges, the model must align positional information from multiple views to construct a coherent understanding of spatial relationships within the scene.

## A.2   Dataset Composition and Statistics

Figure 7 illustrates the statistics of the E3VQA benchmark. Figure 7(a) shows the distribution of correct answer positions among the four options (A-D) for each category. The uniform distribution of correct answers indicates that the dataset is balanced with respect to answer positions. Figure 7(b) shows the distribution of source video types used to construct E3VQA, demonstrating the benchmark's broad coverage of real-world user-interaction scenarios. Finally, Figure 7(c) illustrates the detailed composition of question types within each category, underscoring E3VQA's broad scope of evaluation.

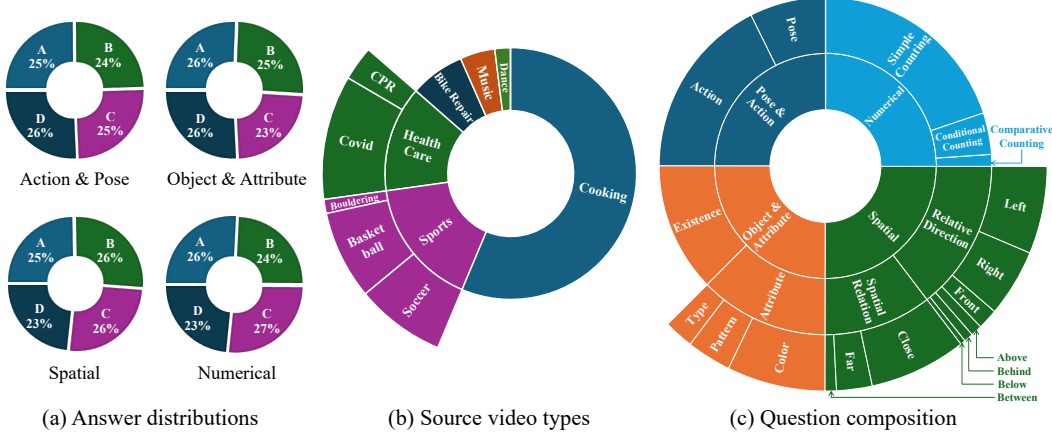

(a) Answer distributions      (b) Source video types      (c) Question composition

Figure 7: E3VQA statistics: (a) Distribution of the correct answers among the four options (A-D), (b) Distribution of source video types used to construct E3VQA, and (c) Composition of question types within each category.

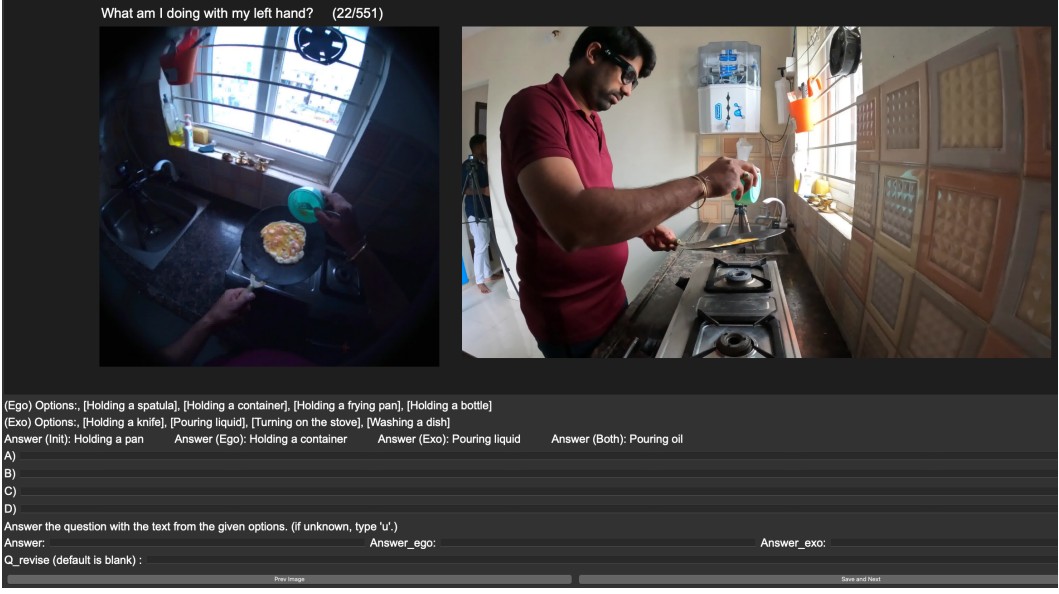

Figure 8: User interface used by annotators during human verification.

## A.3    Details of Human Verification

The human verification is conducted by four members of our research team (the authors), all of whom have domain expertise in vision-language reasoning and extensive experience with ego-exo data. Each annotator is assigned to a specific question category and conducts verification using the user interface shown in Figure 8. Annotators first remove questions that are ambiguous or unanswerable when both ego and exo views are provided. Next, they discard overly simple questions that can be solved through shallow pattern matching, such as cases where $A_{both}$ is correct regardless of reasoning, even if they pass the question filtering stage. Then, annotators utilize the view-specific responses generated in Section 2.3.2 to construct answer options. To supplement potentially redundant or low-quality responses, they additionally use two option sets, each containing four candidate options derived from the ego and exo images, respectively. Using these option sets, the annotators curate complete question sets that include the question, the correct answer, and the distractor options. Finally, each annotator

Table 5: Comparison of open-source LVLMs in terms of architecture (vision encoder and LLM) and use of egocentric data during training.

| Model | Vision Encoder | LLM Backbone | Train w/ Ego Data |
|---|---|---|---|
| InternVL3-14B | InternViT-300M-448px-V2.5 | Qwen2.5-14B | **Not Provided** |
| Qwen2.5-VL-7B | ViT (customized) | Qwen2.5-7B | **Not Provided** |
| Qwen2-VL-7B | ViT-L | Qwen2-7B | ✗ |
| LLaVA-NeXT-OneVision-7B | SigLIP-SO | Qwen2-7B | ✓ |
| InternVL2-8B | InternViT-300M | Qwen2.5-7B | ✓ |
| LLaVA-NeXT-Interleave-7B | SigLIP-SO | Qwen1.5-7B | ✗ |
| MANTIS-Idefics2-8B | SigLIP | Mistral-7B-v0.1 | ✗ |
| Deepseek-VL-chat-7B | SigLIP-L, SAM-B | DeepSeek-LLM-7B | ✗ |
| Qwen-VL-Chat-7B | ViT-bigG | Qwen-7B | ✗ |

Table 6: Performance comparison of recent closed and open-source models on E3VQA in the text-only setting (without visual input).

| LVLMs | Action | | Object | | Numerical | | Spatial | | Avg. |
|---|---|---|---|---|---|---|---|---|---|
| | Ego | Exo | Ego | Exo | Ego | Exo | Ego | Exo | |
| **Closed-Source** | | | | | | | | | |
| Gemini 2.0 Flash | 20.20 | 24.80 | 20.60 | 22.80 | 27.80 | 30.20 | 16.20 | 15.20 | 22.23 |
| GPT-4o | 25.00 | 30.40 | 12.20 | 23.20 | 7.40 | 30.80 | 11.60 | 15.40 | 19.50 |
| **Open-Source** | | | | | | | | | |
| InternVL3-14B | 31.00 | 37.20 | 33.80 | 30.20 | 38.40 | 32.80 | 10.80 | 11.80 | 28.25 |
| Qwen2.5-VL-7B | 32.20 | 34.60 | 32.80 | 31.00 | 35.20 | 36.20 | 8.00 | 5.40 | 26.93 |
| LLaVA-OneVision-7B | 32.60 | 38.00 | 31.40 | 26.80 | 27.40 | 28.80 | 7.00 | 5.60 | 24.70 |

reviews subsets created by others to ensure consistency and clarity across the dataset, filtering out any ambiguous or low-quality instances.

### A.4 Robustness to Temporal Misalignment

To evaluate the robustness of the E3VQA benchmark under temporal misalignment between egocentric and exocentric views, we analyze how synchronization gaps affect the consistency of ground-truth answers. Specifically, we randomly sample 80 examples (10 from each category) and evaluate how the ground truth changes when the ego and exo views are misaligned by 1 to 3 seconds. When the synchronization gap is 1 or 2 seconds, 99% and 97% of the ground-truth answers remain unchanged, respectively. Even with a 3-second gap, 94% of the answers remain consistent. Although maintaining perfect temporal alignment can be challenging in real-world applications, our analysis shows that small temporal mismatches (e.g., a few seconds) have minimal impact on the consistency of the ground truth.

## B Experimental Details

### B.1 LVLMs and Evaluation Setup

Table 5 provides an overview of the LVLMs used in our experiments in terms of the model architecture and the use of egocentric data in training. These models are selected based on their capability of processing multi-image inputs. By evaluating models with diverse vision-language architectures, we examine how recent LVLMs respond to and reason through the challenges posed by the E3VQA benchmark. For evaluation, we use NVIDIA RTX A6000 GPUs. All evaluation results are reported as the mean and standard deviation over three independent runs, using each model's default generation settings.

### B.2 Text-Only Baseline Results

To investigate potential biases in the E3VQA benchmark, we evaluate recent LVLMs under a text-only setting, where only the textual question is provided without any visual input. This experiment allows us to examine whether models can exploit linguistic patterns or biases present in the data without

Table 7: Performance comparison of recent multimodal CoT methods on E3VQA constructed from LEMMA.

| Methods | Pose & Action | | Object & Attribute | | Numerical | | Spatial | | Avg. |
|---|---|---|---|---|---|---|---|---|---|
| | Ego | Exo | Ego | Exo | Ego | Exo | Ego | Exo | |
| **GPT-4o** | | | | | | | | | |
| Default | 39.68 | 43.55 | 63.49 | 72.58 | 56.45 | 46.03 | 39.68 | 45.16 | 50.83 |
| DDCoT [54] | 38.10 | 48.39 | 66.67 | 70.97 | **64.52** | 47.62 | 47.62 | 38.71 | 52.83 |
| CoCoT [50] | 44.44 | 46.77 | 69.84 | 70.97 | 54.84 | 47.62 | 46.03 | **51.61** | 54.02 |
| CCoT [32] | 36.51 | 48.39 | 68.25 | 69.35 | 53.23 | 47.62 | 47.62 | 40.32 | 51.41 |
| **M3CoT (Ours)** | **44.44** | **51.61** | **71.43** | **75.81** | 59.68 | **65.08** | **63.49** | 50.00 | **60.19** |
| **Gemini 2.0 Flash** | | | | | | | | | |
| Default | 39.68 | 53.23 | 66.67 | 74.19 | 51.61 | 46.03 | 46.03 | 46.77 | 53.03 |
| DDCoT [54] | 39.68 | 46.77 | 60.32 | **77.42** | 54.84 | 47.62 | 55.56 | 40.32 | 52.82 |
| CoCoT [50] | **46.03** | 48.39 | 65.08 | 74.19 | 50.00 | 47.62 | 47.62 | 40.32 | 52.41 |
| CCoT [32] | 44.44 | 51.61 | 66.67 | 72.58 | 56.45 | **49.21** | 46.03 | 40.32 | 53.41 |
| **M3CoT (Ours)** | 44.44 | **59.68** | **66.67** | 72.58 | **56.45** | 42.86 | **55.56** | **50.00** | **56.03** |

Table 8: Performance comparison of recent multimodal CoT methods on open-source models.

| Methods | Pose & Action | | Object & Attribute | | Numerical | | Spatial | | Avg. |
|---|---|---|---|---|---|---|---|---|---|
| | Ego | Exo | Ego | Exo | Ego | Exo | Ego | Exo | |
| **InternVL3 - 14B** | | | | | | | | | |
| Default | $44.73_{\pm 1.50}$ | $54.93_{\pm 1.42}$ | $68.13_{\pm 0.81}$ | $73.73_{\pm 0.99}$ | $35.60_{\pm 1.11}$ | $\mathbf{53.00}_{\pm 0.20}$ | $45.67_{\pm 0.58}$ | $48.33_{\pm 0.99}$ | 53.02 |
| DDCoT [54] | $47.87_{\pm 0.83}$ | $58.33_{\pm 2.64}$ | $68.47_{\pm 0.50}$ | $72.67_{\pm 1.42}$ | $35.33_{\pm 2.53}$ | $46.80_{\pm 2.12}$ | $50.67_{\pm 1.10}$ | $45.93_{\pm 0.95}$ | 53.26 |
| CoCoT [50] | $\mathbf{49.53}_{\pm 0.81}$ | $57.27_{\pm 0.50}$ | $68.27_{\pm 1.14}$ | $72.53_{\pm 1.14}$ | $34.87_{\pm 1.55}$ | $47.93_{\pm 0.64}$ | $49.20_{\pm 1.91}$ | $46.27_{\pm 0.95}$ | 53.23 |
| CCoT [32] | $44.60_{\pm 1.91}$ | $58.40_{\pm 2.09}$ | $65.27_{\pm 0.64}$ | $73.80_{\pm 0.53}$ | $\mathbf{37.80}_{\pm 3.30}$ | $50.00_{\pm 0.72}$ | $46.27_{\pm 1.33}$ | $48.80_{\pm 1.78}$ | 53.12 |
| **M3CoT (Ours)** | $45.87_{\pm 1.21}$ | $\mathbf{60.00}_{\pm 0.35}$ | $\mathbf{70.60}_{\pm 0.40}$ | $\mathbf{75.73}_{\pm 0.76}$ | $35.07_{\pm 0.50}$ | $50.87_{\pm 0.70}$ | $\mathbf{50.80}_{\pm 0.92}$ | $\mathbf{49.40}_{\pm 0.72}$ | **54.79** |
| **InternVL3 - 8B** | | | | | | | | | |
| Default | $43.70_{\pm 3.25}$ | $54.90_{\pm 0.42}$ | $64.80_{\pm 0.85}$ | $70.30_{\pm 0.71}$ | $35.90_{\pm 2.12}$ | $45.20_{\pm 1.13}$ | $42.10_{\pm 2.97}$ | $46.60_{\pm 3.11}$ | 50.44 |
| DDCoT [54] | $\mathbf{48.10}_{\pm 0.99}$ | $\mathbf{59.20}_{\pm 4.24}$ | $67.20_{\pm 0.28}$ | $68.80_{\pm 1.13}$ | $34.60_{\pm 0.85}$ | $47.00_{\pm 0.00}$ | $46.30_{\pm 2.69}$ | $45.80_{\pm 1.41}$ | 52.13 |
| CoCoT [50] | $43.90_{\pm 0.14}$ | $58.20_{\pm 1.13}$ | $65.10_{\pm 0.42}$ | $68.40_{\pm 1.13}$ | $37.10_{\pm 1.84}$ | $48.70_{\pm 0.42}$ | $43.50_{\pm 2.97}$ | $43.40_{\pm 0.57}$ | 51.04 |
| CCoT [32] | $44.00_{\pm 0.85}$ | $55.00_{\pm 1.41}$ | $63.60_{\pm 0.57}$ | $68.30_{\pm 1.27}$ | $35.50_{\pm 0.42}$ | $\mathbf{51.20}_{\pm 1.41}$ | $43.40_{\pm 0.57}$ | $44.70_{\pm 3.54}$ | 50.71 |
| **M3CoT (Ours)** | $45.50_{\pm 0.14}$ | $57.20_{\pm 0.00}$ | $\mathbf{68.20}_{\pm 0.00}$ | $\mathbf{71.20}_{\pm 0.00}$ | $\mathbf{37.60}_{\pm 0.00}$ | $47.50_{\pm 0.71}$ | $\mathbf{53.80}_{\pm 0.00}$ | $\mathbf{49.20}_{\pm 0.00}$ | **53.78** |

relying on visual reasoning. The results in Table 6 show that all models exhibit substantially lower performance in the absence of visual information, confirming that the E3VQA benchmark requires ego-exo image-grounded reasoning rather than relying solely on textual cues.

# C Additional Experiments and Analysis of M3CoT

## C.1 Generalization across Datasets

To validate the generalization of our proposed framework across different datasets, we further extend E3VQA using LEMMA [14], a multi-view dataset that captures goal-directed daily activities in home environments. Unlike Ego-Exo4D, the ego images in LEMMA are in a standard rectangular format (i.e., rectified) without the dark peripheral regions. Following the same data generation pipeline in Section 2.3, we construct 500 additional samples evenly distributed across all categories.

As shown in Table 7, the results demonstrate that our E3VQA benchmark construction pipeline generalizes well across datasets, indicating that the difficulty of ego-exo multi-view reasoning arises not from dataset-specific visual properties but from the inherent challenge of integrating complementary perspectives for visual understanding. Furthermore, the consistent improvements achieved by M3CoT across datasets highlight the robustness of its multi-perspective reasoning mechanism.

Table 9: Performance comparison across question subsets grouped by required views.

| LVLMs | Methods | Any | Ego | Exo | Both | Avg. |
|---|---|---|---|---|---|---|
| | Default | 65.20 | 63.96 | 58.59 | 39.28 | 56.76 |
| | DDCoT | 69.71 | 69.37 | 60.83 | 37.68 | 59.40 |
| GPT-4o | CoCoT | 66.94 | 64.41 | 61.47 | 43.48 | 59.08 |
| | CCoT | 67.94 | 67.50 | 61.05 | 42.03 | 59.63 |
| | **M3CoT** | **73.45** | **74.30** | **64.23** | **52.61** | **66.15** |
| | Default | 65.81 | 66.08 | 54.38 | 37.52 | 55.95 |
| | DDCoT | 66.48 | 66.36 | 58.22 | 31.13 | 55.55 |
| Gemini 2.0 Flash | CoCoT | 67.23 | 65.43 | 56.44 | 32.43 | 55.38 |
| | CCoT | 65.76 | 66.18 | 55.29 | 43.21 | 57.61 |
| | **M3CoT** | **69.76** | **69.24** | **62.97** | **53.19** | **63.79** |

## C.2 Evaluation on Open-Source Models

We present experimental results of our M3CoT prompting technique compared to existing CoT methods on open-source LVLMs. Specifically, we apply M3CoT on InternVL3-14B [55], the top-performing open-source model, and further evaluate the performance on InternVL3-8B. As shown in Table 8, most CoT methods result in only marginal performance gains, with several failing to improve accuracy and even causing degradation in certain categories. This aligns with prior findings suggesting that the CoT method is often ineffective in smaller models with limited reasoning capability [41, 19, 7]. Despite the limitations observed in smaller models, our M3CoT consistently achieves superior performance compared to other CoT methods, highlighting its robustness across model sizes.

## C.3 Performance across Question Subsets Based on Required Views

We further compare model performance across the four subsets (Any, Ego, Exo, and Both) introduced in Section 5.2 to examine how our M3CoT approach provides advantages across different question types. Among these, performance on the Both subset reflects a model's ability to integrate multi-view information. Results on the Any subset reflect how well a model handles redundant or overlapping information, while those on the Ego and Exo subsets indicate a model's robustness when unnecessary or irrelevant information is included in the input. As shown in Table 9, M3CoT consistently outperforms other CoT-based methods across all subsets, demonstrating its overall effectiveness in multi-view reasoning.

## C.4 Evaluation of Scene Graph Quality

The quality of scene graphs is often evaluated using the intersection-over-union (IoU) with ground-truth scene graphs [46, 52]. However, ground-truth scene graphs often include overly exhaustive details that may not always be beneficial for answering questions accurately. To better assess the practical utility of scene graphs generated by M3CoT, we introduce two complementary metrics:

- **False Discovery Rate (FDR)** is measured as the proportion of incorrect elements among the predicted scene graph components, including object classes, relations, and attributes. This metric measures how accurately the scene graphs represent the visual scene without including non-existent elements.
- **Answer Accuracy** is measured by model accuracy when the generated scene graph serves as input to the model, indicating how effectively the graph supports reasoning when solving questions.

To evaluate the scene graphs with respect to these two metrics, we constructed a subset of 120 samples from E3VQA, evenly distributed across all question categories. We then evaluated the scene graphs before and after the refinement step of M3CoT. After refinement, the FDR of the generated scene graphs decreased from 9.37% to 6.21%, while the answer accuracy increased from 46.88% to 60.42%. These results demonstrate that the refinement stage of M3CoT not only reduces errors in the scene graphs but also reorganizes their visual information to better support the model in generating accurate answers.

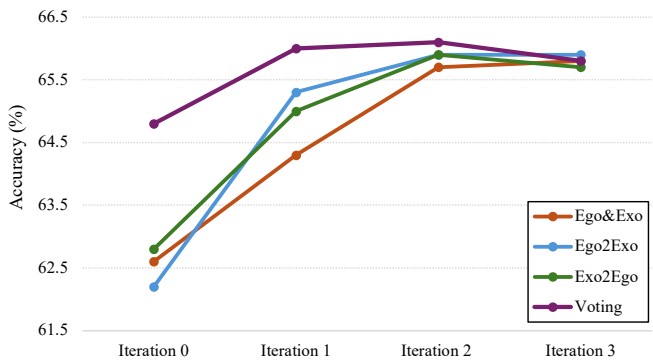

Figure 9: Performance across different perspectives and majority voting results over iteration steps.

Table 10: Comparison of computational overhead between M3CoT and existing CoT prompting methods on GPT-4o and Gemini 2.0 Flash.

| LVLMs | Method | Token Usage | Time (s) | Acc. (%) |
|---|---|---|---|---|
| GPT-4o | Default | 2,120.25 | 2.62 | 60.90 |
| | DDCoT | 6,807.18 | 9.57 | 64.43 |
| | CoCoT | 2,349.23 | 6.25 | 62.87 |
| | CCoT | 9,225.30 | 22.59 | 63.74 |
| | M3CoT | 47,250.12 | 37.56 | 68.58 |
| Gemini 2.0 Flash | Default | 790.60 | 2.56 | 59.80 |
| | DDCoT | 2,752.09 | 5.48 | 60.31 |
| | CoCoT | 950.71 | 3.84 | 61.09 |
| | CCoT | 2,817.97 | 7.14 | 60.18 |
| | M3CoT | 26,164.36 | 20.03 | 66.12 |

## C.5 Analysis of Iteration Steps in Multi-Agent Scene Graph Refinement

To analyze the effect of iteration steps in M3CoT, we report the accuracy of each individual perspective as well as the majority-voted answer derived from them at each iteration. As shown in Figure 9, without any information exchange across perspectives, all individual perspectives and the majority-voted answer achieve relatively low accuracy (iteration 0). As agents begin to exchange their scene graphs, we observe a steady improvement in the accuracy of each individual perspective, suggesting that iterative refinement facilitates mutual enhancement through shared contextual understanding. This process also leads to a corresponding increase in voting accuracy, reflecting not only the enhanced quality of individual predictions but also a stronger consensus across perspectives. However, beyond the second iteration, we find that both individual accuracy and voting accuracy plateau. We attribute this saturation to the convergence of information across agents: while initial iterations benefit from the diversity of complementary perspectives, excessive alignment diminishes the gains from their integration. This observation highlights a trade-off in our multi-perspective refinement strategy between refining individual scene representations and preserving representational diversity. Note that all experiments and analyses in this paper are conducted with a fixed iteration count of 1, using Gemini 2.0 Flash unless otherwise specified.

## C.6 Analysis of Computational Cost

To examine the trade-off between computational efficiency and performance, we measure inference time, token usage, and accuracy of recent CoT prompting methods, including M3CoT, on GPT-4o and Gemini 2.0 Flash (see Table 10). Please note that the reported inference time may be unreliable due to variability in API response, so it should be interpreted as a general trend rather than an exact value. While multi-perspective reasoning requires additional computation, it enables a more accurate understanding than other recent prompting methods.

# D Qualitative Examples of M3CoT

## D.1 Comparison with Other CoT Methods

We provide additional examples that illustrate how M3CoT improves reasoning compared to other CoT approaches (see Figure 10).

## D.2 Scene Graphs from Three Perspectives

To further examine how different perspectives in M3CoT contribute to capturing complementary information, we present additional qualitative examples of scene graphs derived from each perspective. As shown in Figures 11, 12, and 13, the scene graphs from the three perspectives exhibit complementary strengths depending on the question, particularly regarding which image should be referenced to answer it.

## D.3 Conflict Resolution during Refinement

We provide qualitative examples illustrating how M3CoT resolves conflicts among scene graphs from different perspectives during the refinement stage (see Figure 14).

## D.4 Typical Failure Cases

We provide qualitative examples illustrating the failure cases of M3CoT (see Figure 15).

# E Prompt Templates

## E.1 Prompt Templates for E3VQA Construction

To guide LVLMs in understanding the question categories and tasks for generating meaningful question-answer pairs, we carefully design the prompts for each stage. To generate question-answer pairs from a single viewpoint, we use the prompts shown in Figure 16–25. For view-specific response generation, we apply the prompts in Figure 26–33. For response-based filtering, we use the prompts shown in Figure 34 and 35. Finally, to generate four candidate options from either the ego or exo image, we use the prompts illustrated in Figure 36 and 37.

## E.2 Prompt Templates for Baselines and CoT Methods

The system and user prompts used in the baseline experiments of E3VQA are shown in Figure 38. In addition, for M3CoT, the prompts for scene graph generation from each perspective are shown in Figure 39–41, and the prompts for scene graph refinement across agents are presented in Figure 42. The prompts used in other CoT baselines are shown in Figure 43–45.

# F Limitations

Despite its contributions, this work has several limitations. First, the E3VQA benchmark is constructed solely from the Ego-Exo4D and LEMMA datasets, which may exhibit dataset bias and limited generalizability in diverse real-world scenarios. Second, although the queries and answer options in E3VQA are carefully crafted, they may not fully capture the diversity of natural language expressions and user intents encountered in real-world interactions with visual AI assistants. Third, while recent AI APIs offer a solution for scaling the benchmark, their use entails substantial financial costs. Fourth, M3CoT introduces increased computational overhead due to its multi-step reasoning across multiple perspectives, which may limit its applicability in resource-constrained scenarios. Finally, since E3VQA is constructed from images rather than videos, the benchmark may not fully assess an LVLM's ability to capture temporal cues and motion dynamics, an aspect we leave for future work.

# G    Ethics Statement

This work has the potential to positively impact society by enhancing the capabilities of visual assistants and embodied AI systems, particularly in scenarios that require comprehensive scene understanding from both egocentric and exocentric views. Such advancements may enhance human-AI interaction and improve support in assistive technologies. However, the use of egocentric visual data may raise important privacy concerns, especially in sensitive environments. We acknowledge these risks and emphasize the importance of implementing safeguards and transparency mechanisms in future deployments. As part of our commitment to responsible data use, we have obtained the appropriate licenses from the contributing institutions for the use of the Ego-Exo4D dataset in this research.

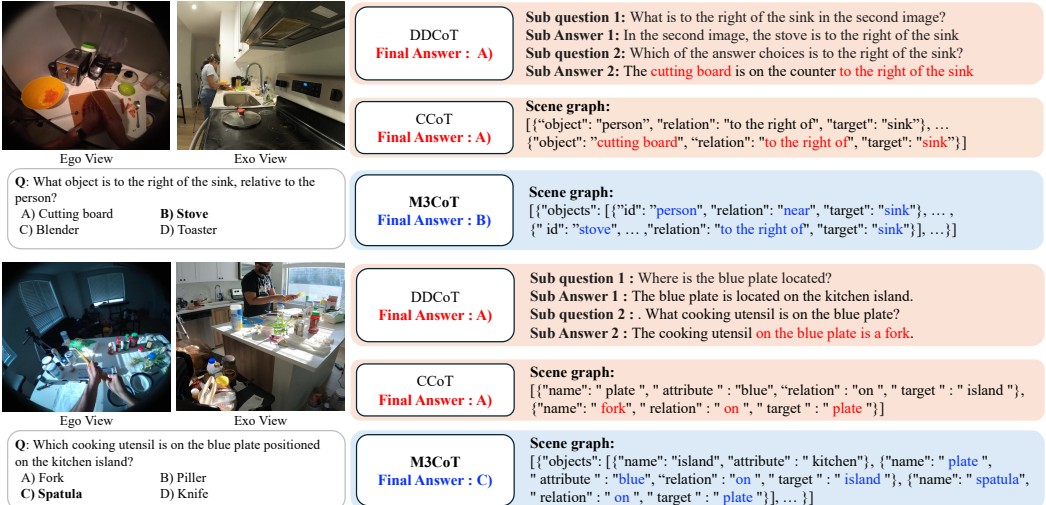

Figure 10: Qualitative examples of answers and reasoning processes generated by different prompting methods.

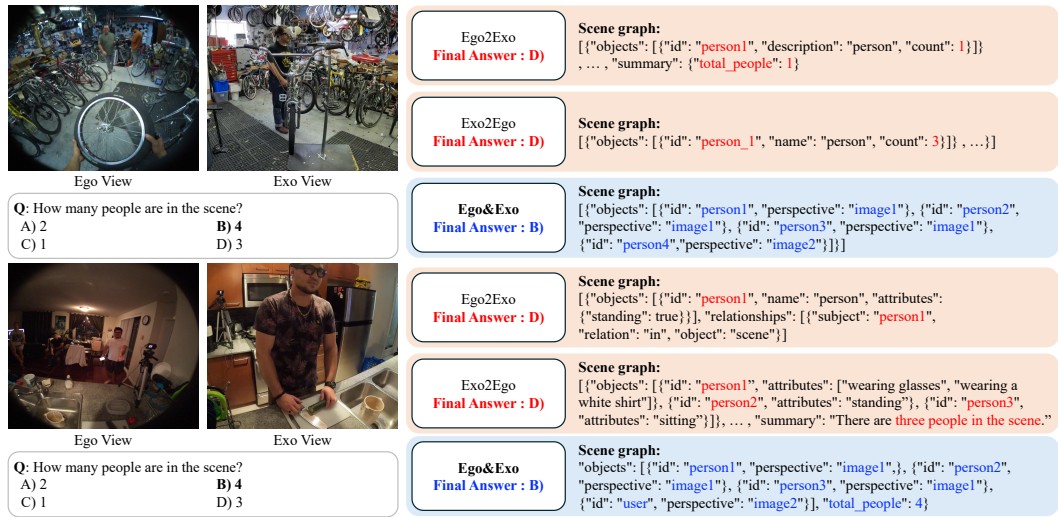

Figure 11: Qualitative examples of answers and reasoning processes generated by different perspectives. The scene graph from the Ego&Exo perspective demonstrates a strong capability to capture the information necessary for answering questions grounded in both ego and exo views.

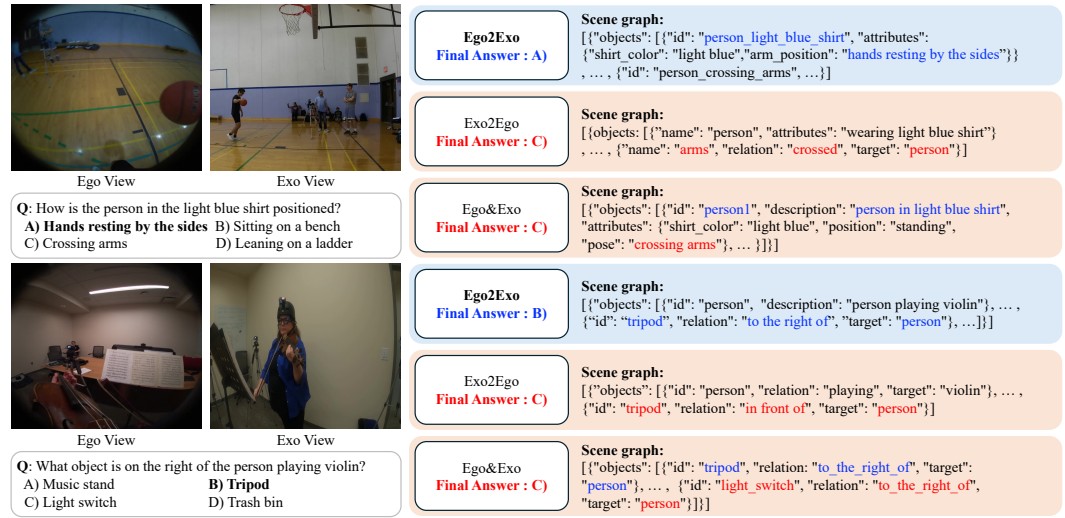

Figure 12: Qualitative examples of answers and reasoning processes generated by different perspectives. The scene graph from the Ego2Exo perspective demonstrates a strong capability to capture the information necessary for answering questions grounded in the exo view alone.

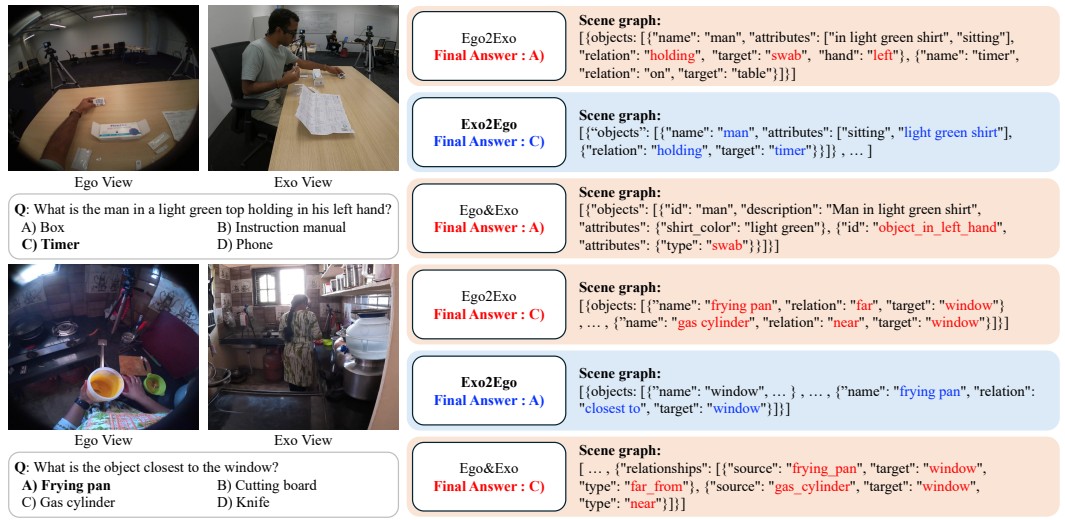

Figure 13: Qualitative examples of answers and reasoning processes generated by different perspectives. The scene graph from the Exo2Ego perspective demonstrates a strong capability to capture the information necessary for answering questions grounded in the ego view alone.

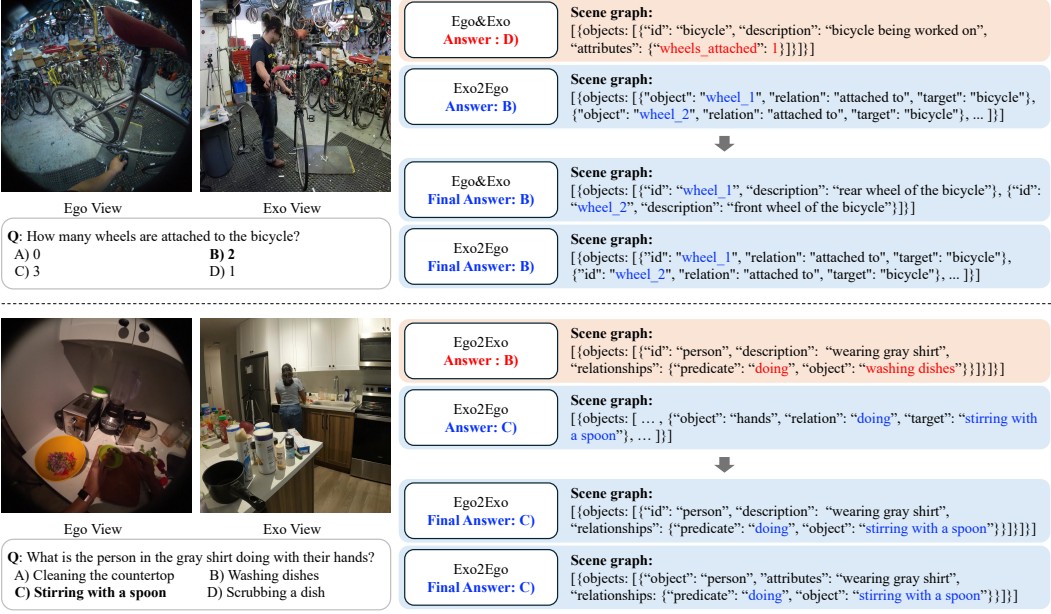

Figure 14: Qualitative examples illustrating how conflicts among scene graphs from different perspectives are resolved by M3CoT. Scene graphs with inaccurate elements are refined during M3CoT's refinement stage, yielding accurate and consistent representations.

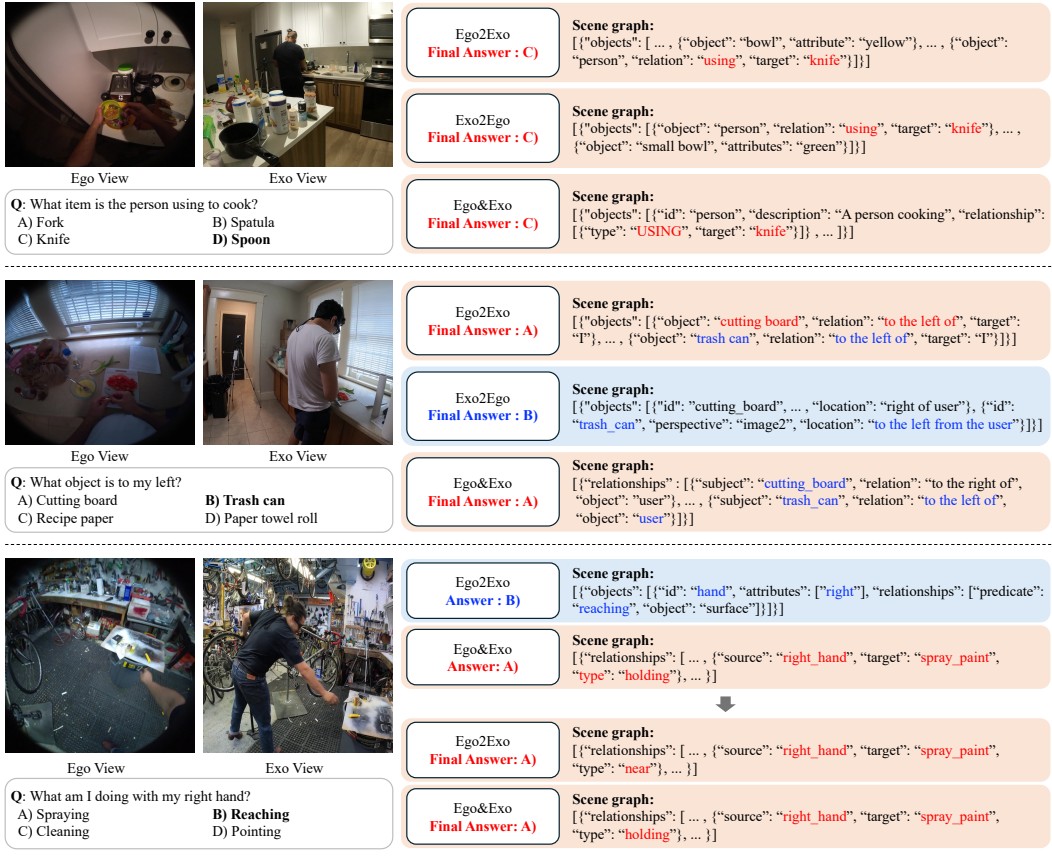

Figure 15: Qualitative examples illustrating M3CoT's failure cases. The top row shows examples where all three perspectives produce incorrect predictions. The middle row illustrates cases where the scene graphs are correctly generated but fail to yield accurate answers. The bottom row shows cases where the refinement stage fails due to erroneous scene graphs.

**Egocentric Single-View QA Generation Prompt**

{Ego Image}

You are given the visual input from the camera worn by the user (referred to as 'I').
Based on this visual input, generate three question-answer pairs.
Ensure that the generated question-answer pairs are directly based on the visual input.

{Category-wise Prompt}

Requirements:
Each question must explicitly include the pronoun 'I' or 'me' to ensure the focus remains on the user.
Each answer should be a single word or a short phrase.
Ensure that all three question-answer pairs meet these criteria and are relevant to the visual input.
Strictly adhere to the format of the provided examples.

Figure 16: Egocentric single-view QA generation prompt.

**Egocentric Single-View QA Generation Prompt: Action & Pose**

Instructions :
Each question must focus on my actions , body posture , or gestures .
The answer must be a verb or verb phrase (e.g. , writing , stretching ,
crossing arms ).
Do not generate QA pairs with overly generic answers like 'standing'
or 'reaching '.

Question Categories & Templates :
Actions (What am I doing?)
− What am I doing?
− What am I doing with my [body part]?

Body Posture (How am I positioned?)
− How is my body positioned?
− How am I sitting / standing / lying?
− What is my posture?

Gestures (What movement am I making?)
− What am I doing with my hands?
− What gesture am I making?
− How am I moving my arms / legs / head?

Examples :
Q: How is my body positioned?
A: Sitting cross−legged

Q: What am I doing with my left hand?
A: Holding a book

Q: What gesture am I making?
A: Waving

Figure 17: Egocentric single-view QA generation prompt: Pose & Action.

**Egocentric Single-View QA Generation Prompt: Object & Attribute**

Instructions:
Each question must focus on identifying a specific object (e.g., mug cup, laptop) or describing an attribute of an object (e.g., navy blue, striped pattern) associated with me.
The answer must be a noun or noun phrase, avoiding overly generic responses such as something or object.

Question Categories & Templates:
Object Identification (What am I interacting with?)
– What am I holding?
– What object is on the table beside me?
– Which item am I picking up?

Object Attributes (What does it look like?)
– What color is the shirt I am wearing?
– What pattern is on my jacket?
– What type of shoes am I wearing?

Examples:
Q: What color is the shirt I am wearing?
A: Navy blue

Q: Which object am I holding in my right hand?
A: A small notebook

Q: What pattern does my sweater have?
A: Checkered pattern

Figure 18: Egocentric single-view QA generation prompt: Object & Attribute.

Instructions:
Each question must focus on the spatial relationships between me and objects in my surroundings.
The answer must be a specific object or location descriptor (e.g., coffee cup, bookshelf, under the table).
Do not generate QA pairs with overly generic answers.

Question Categories & Templates:
Object Proximity (What is closest or farthest?):
– What object is closest to me?
– Which object is the farthest from me?
– What is the nearest object to my [body part]?

Relative Positioning (Where are objects located?)
– What object is to my left/right/front/behind?
– Which object is above/below me?
– Spatial Relations (How are objects arranged?)
– Which object is between me and [another object]?

Examples:
Q: What object is closest to my left hand?
A: Coffee cup

Q: Which object is the farthest from me?
A: Bookshelf

Q: What object is on my right side?
A: Tissue

Figure 19: Egocentric single-view QA generation prompt: Spatial.

**Egocentric Single-View QA Generation Prompt: Numerical**

Instructions:
Each question must focus on numerical reasoning by counting or quantifying specific elements directly related to me.
This may include the number of people, objects, or other countable items present in my surroundings.
The answer must be a numerical value that accurately represents the count of the indicated elements.
Do not generate questions about overly generic objects (e.g., items, objects).
All numerical answers must be within the range of 0 to 5.

Question Categories & Templates:
Counting People (How many people are around me?)
– How many people are in the image excluding me?
– How many individuals are facing the same direction as I am?

Counting Objects (How many things are near or with me?)
– How many [objects] am I holding?
– How many [items] are on the table beside me?

Quantitative Comparisons (How do the numbers compare to what I have?)
– How many more books are on my desk than on the shelf?
– By how much does the number of items in my hands exceed the number on the table?

Examples:
Q: How many people are in the image excluding me?
A: 3

Q: How many more bowls are on my table compared to the table behind me?
A: 2

Q: How many apples am I holding?
A: 3

Figure 20: Egocentric single-view QA generation prompt: Numerical.

**Exocentric Single-View QA Generation Prompt**

{Exo Image}

You are given with the visual input from a fixed-position camera
capturing a scene.
Based on this visual input, generate three question-answer pairs.
Ensure that the generated question-answer pairs are directly based on
 the visual input.

{Category-wise Prompt}

Requirements:
Each answer should be a single word or a short phrase.
Ensure that all three question-answer pairs meet these criteria and
are relevant to the visual input.
Strictly adhere to the format of the provided examples.

Figure 21: Exocentric single-view QA generation prompt.

**Exocentric Single-View QA Generation Prompt: Action & Pose**

Instructions:
Each question must focus on the actions, body posture, or gestures
within the scene.
The answer must be a verb or verb phrase (e.g., writing, stretching,
crossing arms).
Do not generate QA pairs with overly generic answers like 'standing'
or 'reaching'.

Question Categories & Templates:
Actions (What is the person doing?)
- What is the [descriptive] person doing?
- What is the [descriptive] person doing with their [body part]?

Body Posture (How is the person positioned?)
- How is the [descriptive] person positioned?
- What is the posture of the [descriptive] person?

Gestures (What movements is the person making?)
- What kind of gesture is the [descriptive] person making?
- How is the [descriptive] person moving their arms/legs/head?

Examples:
Q: What is the man sitting in the chair doing?
A: Watching a phone

Q: What is the posture of the person wearing a green shirt?
A: Raising one arm

Q: What is the woman in the black jacket doing with their right hand?
A: Holding a book

Figure 22: Exocentric single-view QA generation prompt: Action & Pose.

**Exocentric Single-View QA Generation Prompt: Object & Attribute**

Instructions:
Each question must focus on identifying a specific object in the scene (e.g., 'mug cup', 'laptop') or describing an attribute of an object (e.g., 'navy blue', 'striped pattern').
Questions should reference people or objects by descriptors (e.g., 'the woman in the white top', 'the man with the striped shirt').
The answer must be a noun or noun phrase, avoiding overly generic responses such as 'something' or 'object'.

Question Categories & Templates:
Object Identification (What is present?)
– What is the man with the striped shirt holding?
– What object is placed on the table?
– Which item is the woman wearing a blue top picking up?

Object Attributes (What does it look like?)
– What color is the shirt worn by the man wearing a cap?
– What pattern is on the jacket worn by the woman carrying a handbag?
– What type of shoes is the man standing near the window wearing?

Examples:
Q: What color is the top worn by the woman holding the towel?
A: White

Q: Which object is the man in the black shirt holding in his right hand?
A: Smartphone

Q: What pattern does the sweater worn by the person holding a cup have?
A: Checkered pattern

Figure 23: Exocentric single-view QA generation prompt: Object & Attribute.

**Exocentric Single-View QA Generation Prompt: Spatial**

Instructions:
Each question must explicitly reference an object's or a person's spatial relationship within the scene.
The answer must be a specific object or location descriptor (e.g., scissors, frying pan, under the table).
Do not generate QA pairs with overly generic answers.

Question Categories & Templates:
Object Proximity (What is closest or farthest?)
- Which object is closest to the person wearing [specific item]?
- Which object is the farthest from [reference point]?
- What is the nearest object to [specific location or object]?

Relative Positioning (Where are objects located?)
- What object is to the left/right/front/behind of the man with [specific item]?
- What object is to the left/right/front/behind [reference object]?
- Which object is positioned above/below [reference object]?

Spatial Relations (How are objects arranged?)
- Which object is positioned between [object A] and [object B]?
- What item is placed underneath/inside [object]?
- Which object is located between the two people sitting on the \\ bench?

Examples:
Q: What is the object on the far right of the desk?
A: Scissors

Q: Which cookware is closest to the woman wearing a striped shirt?
A: Frying pan

Q: What object is placed directly in front of the man wearing a cap?
A: Backpack

Q: What object is placed underneath the table?
A: Storage box

Figure 24: Exocentric single-view QA generation prompt: Spatial.

**Exocentric Single-View QA Generation Prompt: Numerical**

Instructions:
Each question must focus on numerical reasoning by counting or quantifying specific elements within the scene.
This may include the number of people, objects, or other countable items present in the image.
The answer must be a numerical value that accurately represents the count of the indicated elements.
Do not generate questions about overly generic objects (e.g., items, objects).
All numerical answers must be within the range of 0 to 5.

Question Categories & Templates:
Counting People (How many are there?)
– How many people are in the scene?
– How many individuals are facing the camera?

Counting Objects (How many things are visible?)
– How many objects is [person descriptor] holding?
– How many items are on the table?

Quantitative Comparisons (How do the numbers compare?)
– How many more books are on the table than on the shelf?
– By how much does the number of items in the man's hands exceed the number on the table?

Examples:
Q: How many people are in the scene?
A: 3

Q: How many objects is the woman in the striped shirt holding?
A: 2

Q: How many oranges are placed on the table?
A: 5

Figure 25: Exocentric single-view QA generation prompt: Numerical.

**View-Specific Response Expansion Prompt: Ego View**

{Ego Image}

You are given a visual input from a camera worn by the user (referred to as 'I') along with a corresponding question.
Based on the visual input, generate the best possible answer.

{Category-wise Prompt}

Requirements:
Each answer option should be a single word or a short phrase.
Follow the provided format strictly.

Q: {Question}

Figure 26: View-specific response expansion prompt: Ego view.

**View-Specific Response Expansion Prompt: Exo View**

{Exo Image}

You are given a visual input from a fixed-position camera capturing a scene along with a corresponding question.
Based on the visual input, generate the best possible answer.

{Category-wise Prompt}

Requirements:
Each answer should be a single word or a short phrase.
Follow the provided format strictly.

Q: {Question}

Figure 27: View-specific response expansion prompt: Exo view.

**View-Specific Response Expansion Prompt: Both Views**

{Ego Image}
{Exo Image}

You are provided with two visual inputs in sequence, each captured from a different perspective:
1. The view from the camera worn by the user ('I').
2. The view captured by an external camera observing the user ('I').
These two images capture the same event at the same time.
Based on the visual inputs, generate the best possible answer.

{Category-wise Prompt}

Requirements:
Each answer should be a single word or a short phrase.
Follow the provided format strictly.

Q: {Question}

Figure 28: View-specific response expansion prompt: Both views.

**View-Specific Response Expansion Prompt: Text Only**

Based on the question, generate the best possible answer.

{Category-wise Prompt}

Requirements:
Each answer should be a single word or a short phrase.
Follow the provided format strictly.

Q: {Question}

Figure 29: View-specific response expansion prompt: text only.

**View-Specific Response Expansion Prompt: Action & Pose**

Instructions:
The answer must be a verb or verb phrase (e.g., writing, stretching, crossing arms).
Do not generate overly generic answers like 'standing' or 'reaching'.

Output format:
Q: How is my body positioned?
A: Sitting cross-legged

Q: What is the man sitting in the chair doing?
A: Watching a phone

Figure 30: View-specific response expansion prompt: Action & Pose.

**View-Specific Response Expansion Prompt: Object & Attribute**

```
Instructions:
The answer must be a noun or noun phrase, avoiding overly generic
responses such as 'something' or 'object'.

Output format:
Q: What color is the shirt I am wearing?
A: Navy blue

Q: What color is the top worn by the woman holding the towel?
A: White
```

Figure 31: View-specific response expansion prompt: Object & Attribute.

**View-Specific Response Expansion Prompt: Spatial**

```
Instructions:
The answer must be a specific object or location descriptor (e.g.,
coffee cup, bookshelf, under the table).
Do not generate overly generic answers.

Output format:
Q: What object is closest to my left hand?
A: Coffee cup

Q: What is the object on the far right of the desk?
A: Scissors
```

Figure 32: View-specific response expansion prompt: Spatial.

**View-Specific Response Expansion Prompt: Numerical**

```
Instructions:
The answer must be a numerical value that accurately represents the
count of the indicated elements.
All numerical answers must be within the range of 0 to 5.

Output format:
Q: How many people are in the image excluding me?
A: 3

Q: How many people are in the scene?
A: 3
```

Figure 33: View-specific response expansion prompt: Numerical.

**Response-Based Question Filtering Prompt 1**

```
Here is the question: '{Question}'.

The provided answer is {answer_both}, and the given label is {answer_init}.
Do they convey the same meaning based on the question? Respond with a
  single word or phrase.
```

Figure 34: Response-based question filtering prompt (1).

**Response-Based Question Filtering Prompt 2**

```
Here is the question: '{Question}'.

The provided answer is '{answer_text}', and the given label is
 '{answer_init}'.
Do they convey the same meaning based on the question? Respond with a
  single word or phrase.
```

Figure 35: Response-based question filtering prompt (2).

**Option Generation Prompt: Ego**

```
{Ego Image}
You are given a visual input from a camera worn by the user (referred
  to as 'I').
Based on the following question and answer, generate four multiple-
choice options.
Question: {Question}
Answer: {answer_ego}

Ensure that each incorrect option is closely related to the visual
content, making it challenging to easily identify the correct answer.
Follow the format below exactly:

Options:
[Option1]
[Option2]
[Option3]
[Option4]
```

Figure 36: Option generation prompt: Ego.

**Option Generation Prompt: Exo**

{Exo Image}
You are given a visual input from a fixed-position camera capturing a
 scene.
Based on the following question and answer, generate four multiple-
choice options.
Question: {Question}
Answer: {answer_exo}

Ensure that each incorrect option is closely related to the visual
content, making it challenging to easily identify the correct answer.
Follow the format below exactly:

Options:
[Option1]
[Option2]
[Option3]
[Option4]

Figure 37: Option generation prompt: Exo.

**System Prompt & Question (Instruction) Prompt**

**System Prompt**

You are a helpful assistant.
You are provided with two visual inputs in sequence, each captured
from a different perspective:
1. The view from the camera worn by the user ('I').
2. The view captured by an external camera observing the user ('I').

The first image shows what the user ('I') sees from their perspective
.
The user's full body cannot be visible; you may only see parts of
their body, like their hand, foot, or arm, or in some cases, none of
the user's body at all.

The second image shows both the user and the environment from a third
-person perspective with a broad view.
The user's full body is visible, but due to the fixed viewpoint, some
parts may not be visible.

These two images capture the same event at the same time.
Your task is to analyze both images along with the question and
provide the most accurate response based on the visual information
from both perspectives.

**Question (Instruction) Prompt**

{Ego Image}
{Exo Image}
{Question}

Only one option is correct.
Present the answer in the form X).

Figure 38: System Prompt and Question(Instruction) Prompt.

**M3CoT Prompts - Ego2Exo Perspective**

**Scene graph generation phase (Ego2Exo)**

Task :
For the provided image and its associated question , generate a scene graph in JSON format that includes the following :
1. Objects that are relevant to answering the question .
2. Object attributes that are relevant to answering the question .
3. Object relationships that are relevant to answering the question .

Just generate the scene graph in JSON format . Do not say extra words .

{Ego Image}
{Question Prompt}

---

**Scene graph refinement phase (Ego2Exo)**

Task :
For the provided image from a different view and the scene graph generated from the previous view , refine the scene graph in JSON format as follows :
1. Review and Update Existing Objects and Relationships :
Examine the objects and relationships in the initial scene graph . Update their attributes or positions based on observations from both views . Remove only elements that are clearly erroneous (e.g. , annotation errors or duplicates ) .

2. Incorporate New Information :
Identify and add any new objects or relationships that appear in the new view .

3. Align and Reconcile Across Views :
For overlapping objects and relationships , align them using spatial proximity and semantic similarity . If attribute discrepancies arise , select values that best reflect the combined observations .

Ensure that the updated scene graph is logically and physically consistent , avoiding contradictions or impossible configurations . Just generate the refined scene graph in JSON format . Do not say extra words .

{Exo Image}
{Question Prompt}
{Assistant's response(Ego-only SG)}

---

**Initial question response phase (Ego2Exo)**

Use the images and the refined scene graph as context and answer the following question .

{Ego Image}
{Exo Image}
{Question Prompt}
{Assistant's response(Refined SG)}

Figure 39: M3CoT prompt (1).

## M3CoT Prompts - Exo2Ego Perspective

**Scene graph generation phase (Exo2Ego)**

Task:
For the provided image and its associated question, generate a scene graph in JSON format that includes the following:
1. Objects that are relevant to answering the question.
2. Object attributes that are relevant to answering the question.
3. Object relationships that are relevant to answering the question.

Just generate the scene graph in JSON format. Do not say extra words.

{Ego Image}
{Question Prompt}

---

**Scene graph refinement phase (Exo2Ego)**

Task:
For the provided image from a different view and the scene graph generated from the previous view, refine the scene graph in JSON format as follows:
1. Review and Update Existing Objects and Relationships:
Examine the objects and relationships in the initial scene graph. Update their attributes or positions based on observations from both views. Remove only elements that are clearly erroneous (e.g., annotation errors or duplicates).

2. Incorporate New Information:
Identify and add any new objects or relationships that appear in the new view.

3. Align and Reconcile Across Views:
For overlapping objects and relationships, align them using spatial proximity and semantic similarity. If attribute discrepancies arise, select values that best reflect the combined observations.

Ensure that the updated scene graph is logically and physically consistent, avoiding contradictions or impossible configurations. Just generate the refined scene graph in JSON format. Do not say extra words.

{Exo Image}
{Question Prompt}
{Assistant's response(Exo-only SG)}

---

**Initial question response phase (Exo2Ego)**

Use the images and the refined scene graph as context and answer the following question.

{Ego Image}
{Exo Image}
{Question Prompt}
{Assistant's response(Refined SG)}

Figure 40: M3CoT prompt (2).

**M3CoT Prompts - Ego&Exo Perspective**

**Scene graph generation phase (Ego&Exo)**

Task :
Using the provided two images and their associated question , generate a unified scene graph in JSON format that includes the following :

1. Objects that are relevant to answering the question .
2. Object attributes that are relevant to answering the question .
3. Object relationships that are relevant to answering the question .
4. Ensure that objects and relationships from both perspectives are appropriately aligned , integrated , and refined to provide a complete scene representation .

Just generate the unified scene graph in JSON format . Do not say extra words .

{Ego Image}
{Exo Image}
{Question Prompt}

---

**Initial question response phase (Ego&Exo)**

Use the images and the unified scene graph as context and answer the following question .

{Ego Image}
{Exo Image}
{Question Prompt}
{Assistant's Response(Ego&Exo SG)}

Figure 41: M3CoT prompt (3).

**M3COT Prompts - SG Refinement between Agents**

**Scene graph cross-refinement phase (**Ego&Exo /Ego2Exo /Exo2Ego**)**

Task:
Below are different scene graphs generated using different reasoning methods:

One scene graph: {Ego2Exo SG} / {Exo2Ego SG} / {Exo&Ego SG}
One scene graph: {Exo2Ego SG} / {Exo&Ego SG} / {Ego2Exo SG}

Using the scene graphs generated from different methods as additional context, generate a refined scene graph in JSON format for the provided images and their associated question as follows:

1. Review the objects and relationships from the scene graphs and make any necessary adjustments to better align with both views.
2. Ensure that overlapping objects or relationships between the two views are appropriately aligned and refined, enhancing the accuracy of the scene graph.

Just generate the refined scene graph in JSON format. Do not say extra words.

{Ego Image}
{Exo Image}
{Question Prompt}

---

**Question response phase (**Ego&Exo /Ego2Exo /Exo2Ego**)**

Use the images and the unified scene graph as context and answer the following question:

{Ego Image}
{Exo Image}
{Question Prompt}
{Assistant's response(Unified SG)}

Figure 42: M3CoT prompt (4).

**Other CoT Prompts - DDCoT**

For the provided images and their associated question, think step-by-step about the preliminary knowledge required to answer the question. Deconstruct the problem as completely as possible into necessary sub-questions.

Then, with the aim of helping humans answer the original question, attempt to answer those sub-questions.

The expected answering format is as follows:

Sub-questions:
1. <sub-question 1>
2. <sub-question 2>
...

Sub-answers:
1. <sub-answer 1>
2. <sub-answer 2>
...

{Question Prompt}

---

Context: {Assistant's response}

Give your answer to the question according to the sub-questions and sub-answers.

{Question Prompt}

Figure 43: DDCoT Prompt.

**Other CoT Prompts - CoCoT**

Please tell me the similarities and differences of these two images, and answer to the question.

{Question Prompt}

Figure 44: CoCoT Prompt.

**Other CoT Prompts - CCoT**

For the provided images and their associated question, generate a scene graph in JSON format that includes the following:

1. Objects that are relevant to answering the question.
2. Object attributes that are relevant to answering the question.
3. Object relationships that are relevant to answering the question.

Just generate the scene graph in JSON format. Do not say extra words.

{Question Prompt}

---

Scene Graph: {Assistant's response}

Use the images and scene graph as context and answer the following question.

{Question Prompt}

Figure 45: CCoT Prompt.

