# OpenReview forum: "Towards Comprehensive Scene Understanding: Integrating First and Third-Person Views for LVLMs"
_NeurIPS.cc/2025/Conference — NeurIPS 2025 spotlight_

### Official Review · Reviewer_9jna · 2025-06-29

**Clarity:** 3
**Significance:** 2
**Originality:** 3
**Rating:** 5
**Confidence:** 4

**Summary:**

This a benchmark and analysis paper, focusing on evaluating egocentric (Ego) and exocentric (Exo) reasoning in frontier VLMs, and introduces a dataset derived from the prior EgoExo4D work.

The paper introduces E3VQA, a dataset containing 4K pairs of questions and answers related to ego-exo images. Each question is crafted so that it is difficult to answer using only text or a single image; instead, integrating information from both views is essential to select the correct answer.

The authors also introduce a prompting technique called M3CoT, which uses three agent VLMs that focus on both the ego and exo views, respectively, to extract information from the input views, form scene graphs, and use majority voting to select the best answer.

The E3VQA benchmark demonstrates that frontier VLMs still struggle with cross-view reasoning between ego and exo perspectives. The M3CoT prompting technique improves these models' performance by about 5% on the benchmark. Overall, the paper highlights the limitations of current VLMs in multi-view ego-exo reasoning, analyzes the cases where these models struggle most, and proposes potential improvements.

**Questions:**

The paper is well written and convey the message clearly, I don't have questions regards the paper.
The questions I have is more about the focus to improve the significance of the paper, but may out reach of the scope of this paper:

1. This is unclear that the VLMs performance drop in Ego views compared to Exo views, is not because the uniquce image format or charateristics of the EgoExo4D data. But is the narrowed view sight or context imformation in the ego view.
2. Is the Ego-Exo has its unique challenges compared to Ego-Ego, Exo-Exo settigs? As the paper lacks the comparison between how Ego-Exo compared to Exo-Exo in the same timestamp views in EgoExo4D dataset.
3. The analysis of VLM failures in Ego-Exo settings lacks significance. It remains unclear whether these failures stem from substantial visual differences in the appearance of the same object across views, or if powerful visual encoders still produce similar visual embeddings. Furthermore, it is not clear whether VLMs are establishing visual correspondences between ego and exo views, or if they are primarily relying on extracting language descriptions (as suggested by M3CoT) and matching correspondences in the textual space.

**Ethical Concerns:**

["NO or VERY MINOR ethics concerns only"]

**Final Justification:**

My initial major concern was the unique fisheye lens format of the egocentric view images. The rebuttal clarified that LEMMA uses standard rectangular images for the ego view, and the observed trends are consistent across both LEMMA and EgoExo4D datasets.
Thus I would like to improve my previous rating to acceptance.

**Limitations:**

yes

**Paper Formatting Concerns:**

No, the paper is well formatted.

**Quality:**

3

**Strengths And Weaknesses:**

### Strengths

1. The paper is clearly written and effectively conveys that frontier VLMs have limitations in ego-exo multiview reasoning. The proposed prompting technique is simple yet effective, requiring no additional training.
2. The E3VQA QA pairs are mostly auto-generated and filtered using LLMs and VLMs. The pipeline is clearly described and should be easy to scale as more GT Ego-Exo frames and annotations become available.
3. The benchmark is filtered based on the consistency of model responses when given different access to the views (both, ego, exo, or only text). This process ensures that the questions included truly require reasoning over both Ego and Exo views, rather than allowing models to rely on shortcuts or general knowledge to answer.
4. The simple yet effective M3CoT prompting technique improves Ego and Exo reasoning during inference, and its effectiveness is demonstrated across different VLM models.

### Weaknesses

1. As stated in the paper, the benchmark data is curated solely from EgoExo4D video frames and annotations, which limits the diversity of data sources and may restrict the generalizability of the results and findings.
2. The paper does not sufficiently discuss the unique challenges posed by Ego-Exo view integration compared to Exo or Exo multiview scenarios.
3. It is unclear whether the main challenge for frontier VLMs stems from the inherently unique egocentric viewpoint, or from the specific characteristics of the EgoExo4D ego view format (such as its FOV and the dark areas outside the viewpoint), which may differ significantly from the images typically seen during VLM training. If the egocentric images were calibrated and rectified to resemble more conventional photographs, would the observed difficulty for VLMs persist?
4. The analysis section primarily focuses on the fairness of the benchmark but lacks a analysis of the typical failure patterns of current VLMs and a deeper investigation into why these models struggle with ego and exo settings. For instance, visualizations such as attention maps between ego and exo views could help illustrate how models interpret the relationship between perspectives and whether their focus aligns with the question. If not, identifying and analyzing these failure patterns would provide valuable insights into the limitations of VLMs in this context.

Overall, the quality and clarity of the paper are good. However, due to the aforementioned limitations, I rate the significance of the paper as fair.

---

> ### Author Rebuttal · Authors · 2025-07-31
>
> We appreciate the reviewer’s invaluable comments and feedback on our paper. We tried our best to address each of your questions.
>
> **Weakness** **1:** E3VQA is curated solely from EgoExo4D video frames and annotations, which limits the diversity.
>
>
> **Response:** We understand the reviewer’s concern that relying only on the EgoExo4D dataset may not be sufficient to capture diverse real-world scenarios. In fact, to improve diversity, we extended E3VQA using LEMMA, which is a multi-view dataset focused on goal-directed daily tasks in home environments. Following the same data generation pipeline in this work, we constructed 500 new samples, evenly distributed across all categories. From experimental results on this new subset, we show that our E3VQA benchmark construction pipeline works well across different datasets, and the proposed M3CoT method provides a consistent performance gain (see table below).
>
>
>
>
> | LVLMs  | Method   |   |  Action  |          |        |  Object |         |        | Num.  |         |        | Spatial |         |           | Avg.  |
> |-----------|:------:|-|------------|:-------|-------|---------|:-------|-------|--------|:-------|-------|--------|:-------|---------|-------:|
> |              |          |      | Ego      | Exo    | | Ego         | Exo  |   | Ego        | Exo   |  | Ego        | Exo   |  |    |
> | GPT-4o  | Default  || 39.68     | 43.55   | | 63.49        | 72.58 |   | 56.45       | 46.03|    | 39.68       | 45.16   | | 50.83 |
> | | DDCoT || 38.10     | 48.39   | | 66.67        | 70.97   | | **64.52**       | 47.62   | | 47.62       | 38.71   | | 52.83 |
> | | CoCoT || 44.44     | 46.77   | | 69.84        | 70.97   | | 54.84       | 47.62  |  | 46.03       | **51.61**   | | 54.02 |
> | | CCoT  || 36.51     | 48.39   | | 68.25        | 69.35   | | 53.23       | 47.62  |  | 47.62       | 40.32   | | 51.41 |
> | | **M3CoT**  || **44.44**    | **51.61**  |  | **71.43**        | **75.81**  |  | 59.68       | **65.08** |    | **63.49**       | 50.00  |  | **60.19** |
> ||||
> | Gemini 2.0   | Default  ||  39.68  | 53.23   ||66.67    |74.19|   |51.61    |46.03  |  |46.03    |46.77   | |  53.03  |
> | | DDCoT  ||   39.68 |  46.77 | | 60.32   |**77.42**  | |54.84    |47.62   | |55.56    | 40.32 |  | 52.82   |
> | | CoCoT  ||  **46.03** |  48.39 | | 65.08   | 74.19 | | 50.00   |47.62  |  |47.62    |40.32    ||52.41    |
> | | CCoT   ||   44.44 |  51.61  | |66.67   |72.58  | |56.45    |**49.21**   | |46.03    |40.32  |  |53.41    |
> | | **M3CoT**   ||  44.44  |**59.68**   | |**66.67**    |72.58 |  |**56.45**    |42.86    ||**55.56**    |**50.00**  |  | **56.03**   |
>
>
> **Weakness** **2** **&** **Question** **2:** The paper does not sufficiently discuss the unique challenges posed by Ego-Exo view integration compared to Ego or Exo multiview scenarios.
>
> **Response:** Thank you for raising the important point here. When compared to Ego-Ego or Exo-Exo multi-view scenarios, the Ego-Exo setting presents a unique challenge: answering egocentric questions (e.g., What am I doing?) without explicit information about the identity of “I”. Specifically, when presented with such questions, the model must first infer who “I” refers to, typically by identifying the camera wearer in the egocentric view, and then locate the same person in the exocentric view. This cross-view grounding requires additional reasoning that is not needed in the Ego-Ego or Exo-Exo settings, as “I” cannot be identified in those settings without extra context or annotations. We will incorporate this discussion into the revised manuscript.
>
>
> **Weakness** **3** **&** **Question** **1:** It is unclear whether the main challenge for frontier VLMs stems from the inherently unique egocentric viewpoint, or from the specific characteristics of the EgoExo4D ego view format.
>
> **Response:** As we mentioned, we additionally constructed a benchmark using the LEMMA dataset. Unlike EgoExo4D, the ego images in LEMMA are in a standard rectangular format (i.e., rectified) and do not exhibit dark peripheral regions along the edges. While the ego images exhibit a standard rectangular format, the experimental results for both the baseline and M3CoT show trends consistent with those observed on the E3VQA benchmark generated from EgoExo4D. This consistency implies that the challenge does not arise merely from the specific visual characteristics of EgoExo4D, but rather from the inherent difficulty of integrating egocentric and exocentric views for visual reasoning.
>
> **Weakness** **4** **&** **Question** **3:** The analysis section lacks an analysis of the typical failure patterns of current VLMs.
>
>
> **Response:** To analyze the failure patterns in Ego-Exo multi-view settings, we conducted an experiment in Section 4.1 and Figure 5(c), examining how model performance varies when provided with one view versus both views. We find that, in the ‘Ego’ and ‘Exo’ subsets where the answer-relevant information is present in only one view, providing both views to the model leads to worse performance compared to providing only the informative view. This reveals a failure pattern in which the additional, uninformative view distracts or confuses the model.
>
> While this analysis provides some insights, we acknowledge that it does not fully reveal the fundamental reasons why current LVLMs struggle in Ego-Exo multi-view scenarios. We have tried to interpret the model’s behavior by analyzing attention maps from the decoder layers of the LVLM. However, we were unable to identify consistent and meaningful patterns. Some examples produced correct answers despite weak attention to regions containing the relevant information, while others failed even with seemingly appropriate focus on those regions. We suspect that this may be due to multiple factors, including the lack of sufficient ego-exo paired training data with strong multi-view alignment signals.
>
> We assume that guiding the model through step-by-step reasoning, as done in M3CoT, is beneficial because it is inherently difficult for the model to simultaneously establish correspondences across views and link them to the user’s query in a single step. We believe that M3CoT unlocks the LVLM’s inherent ability to build such correspondences to some extent. We will include all analyses and visualizations in the revised manuscript. We hope our responses can address your concerns.

---

> > ### Author Response · Authors · 2025-08-07
> >
> > Thank you once again for taking the time to review our paper and for providing valuable comments.
> > Following your feedback, we have sincerely worked to address your concerns.
> >
> > As the discussion deadline approaches, please don’t hesitate to let us know if you have any further questions. We would be glad to respond and are committed to revising the paper further to reflect your suggestions and enhance its quality. Thank you for your time and thoughtful consideration.

---

> > > ### Comment · Reviewer_9jna · 2025-08-07
> > >
> > > Thank you authors for the detailed response. My initial major concern was the unique fisheye lens format of the egocentric view images. The rebuttal clarified that LEMMA uses standard rectangular images for the ego view, and the observed trends are consistent across both LEMMA and EgoExo4D datasets. My other questions and concerns have been addressed.
> > >
> > > The benchmark highlights deeper challenges related to uninformative views that may distract or confuse models when reasoning between first and third person perspectives, and these challenges are not attributed to image formats. Therefore, I would like to update my rating to Accept.

---

> > > > ### Author Response · Authors · 2025-08-08
> > > >
> > > > Thank you very much for your thoughtful feedback and for taking the time to review our work. Your suggestions have been helpful in improving the quality of our study, and we appreciate you recognizing our contributions.
> > > >
> > > > We are committed to thoroughly addressing your comments and faithfully implementing your recommendations in the revised manuscript.

---

### Official Review · Reviewer_Gcmm · 2025-06-30

**Clarity:** 3
**Significance:** 3
**Originality:** 3
**Rating:** 5
**Confidence:** 4

**Summary:**

This paper studies the importance of integrating first and third-person views for Vision-and-Language models. This is an incredibly relevant setup particularly for assisted living scenario where the personal assistant might have access to multiple cameras that are installed in a smart home. To tackle this problem, the authors automatically generate a VQA dataset from EgoExo4D. The output of the automatic data collection is then verified by four expert annotators, who were able to produce 4,000 high-quality examples.

Given this dataset, the authors complete an evaluation of both closed-source and open-weights models. Showcasing interesting results which highlight that GPT4o is the top-performing model overall with InternVL3-14B leading the way for open-weights models. The authors also present M3CoT, a prompting technique that uses scene graphs to prompt the model with different information that are used to help the model predict the right answer. This involves multiple steps of scene graph extraction and refinement. The authors evaluate this technique for Gemini and OpenAI models, showcasing how it helps improve performance over other prompting techniques.

**Questions:**

Overall, I really enjoyed the paper and I hope that the authors find my weaknesses informative to help them shape the next iteration of this paper. I'll report below the question I had which I hope the authors will help me improve my understanding of this paper:

1. Who are the expert annotators? What was the procedure they followed to annotate the data?
2. What is the computational costs of running your M3CoT compared to other state-of-the-art methods as well as compared to the baseline? I think it's important to be transparent about this, considering that you are running multiple copies of the same model. I understand that you applied this only to API models therefore you didn't consider the scenario where the user has to deploy this.
3. Related to the previous point, why did you apply the M3CoT only to frontier models? Was there anything that prevented you from applying it to other models like QwenVL?
4. I think it would be great if your table of results could always report the "both" setting because this will highlight a more realistic scenario which is where the model has to truly use both perspectives to answer the question

**Ethical Concerns:**

["NO or VERY MINOR ethics concerns only"]

**Final Justification:**

The authors provided some important details that were not reported in the original paper. I think these additions will make the paper stronger and I hope that the authors will include all the suggested revisions. My score remains the same and I would be happy to see this paper accepted.

**Limitations:**

yes

**Quality:**

3

**Strengths And Weaknesses:**

### Strengths

1. The paper is well-written and easy to follow. The narrative is clear, and the main methodology as well as the results are well described.
2. High-quality data collection defined to derive the final dataset. This represents an important resource for the community and I hope that the authors will release both the dataset as well as their data generation procedure.
3. Interesting prompting technique proposed however I am a bit skeptical of its ability to perform at scale (see weaknesses)
4. Really comprehensive evaluation of several closed and open models. Additionally, the authors provide a detailed error analysis that I find really informative.

### Weaknesses

1. The paper does not feature a specific related work section. I understand that this paper represents a very novel contribution however I wonder whether it would be useful to include a description of the different egocentric question answering datasets such as OpenEQA, and describe how this dataset is different from others. This could be not only in terms of the different perspectives available but also in terms of the different linguistic properties and capabilities required by each question

2. I think it would have been useful to also include some text-only baselines as well to understand whether there are inherent biases in the data.

3. I really liked this paper because I think that the automated data collection pipeline is very interesting and is designed to ensure high-quality annotations. However, I don't find the scene graph prompting technique as a strong contribution of this paper. Specifically, I wonder how scalable this solution is, especially if we imagine a scenario where the model has to react in real-time because it has to process the video stream coming from some smart glasses (e.g., Rayban Glasses). Running multiple copies of the model for each query sounds extremely expensive. It would be good if the authors could provide more details regarding this.

---

> ### Author Rebuttal · Authors · 2025-07-31
>
> We appreciate the reviewer’s invaluable comments and feedback on our paper. We tried our best to address each of your questions.
>
>
> **Weakness** **1:** The paper does not feature a specific related work section.
>
> **Response:** We would like to politely point out that Table 4 and Appendix A.2 in the supplementary material provide a detailed comparison between E3VQA and existing VQA datasets, including OpenEQA.
> To summarize, EgoVQA evaluates the understanding of egocentric visual information. EgoSchema extends this line of work by focusing on long-term reasoning over egocentric experiences. EgoThink emphasizes first-person cognition through questions grounded in the user's perspective, involving the pronoun “I”. EmbodiedQA and OpenEQA also utilize egocentric visual input, but place more emphasis on environment-level understanding, rather than the user’s state or interactions.
> Notably, E3VQA not only leverages multiple visual perspectives (ego and exo) but also includes a broad range of questions targeting the user, the environment, and their interactions, using both first-person and third-person question forms. In particular, it uniquely evaluates a model’s ability to integrate complementary information across ego and exo views, which distinguishes it from prior benchmarks.
>
>
> **Weakness** **2:** I think it would have been useful to also include some text-only baselines as well to understand whether there are inherent biases in the data.
>
> **Response:** We appreciate the reviewer’s suggestion that text-only baselines can be a helpful way to examine potential biases in the data. To address the reviewer’s suggestion, we conducted experiments using text-only inputs (i.e., questions without images), and the results are shown in the table below. (These results will be included in the final manuscript.)
>
> | LVLMs|Action|||  Object ||| Num.  ||| Spatial ||| Avg.  |
> |-----------------|------------|:-------|-------|---------|:-------|-------|--------|:-------|-------|--------|:-------|---------|-------:|
> || Ego| Exo|| Ego| Exo || Ego| Exo|| Ego  | Exo   |||
> | **Closed-Source**|
> | Gemini 2.0| 20.20| 24.80||20.60| 22.80  | |27.80  | 30.20  || 16.20    | 15.20  || 22.23 |
> | GPT-4o| 25.00   | 30.40  || 12.20| 23.20  || 7.40   | 30.80  || 11.60    | 15.40  || 19.50 |
> | **Open-Source**|
> | InternVL3-14B| 31.00| 37.20  ||33.80   | 30.20  || 38.40  | 32.80  || 10.80    | 11.80  || 28.25 |
> | Qwen2.5-VL-7B| 32.20   |34.60   || 32.80   | 31.00  || 35.20 | 36.20  || 8.00     | 5.400   || 26.93 |
> | LLaVA-OneVision-7B | 32.60 | 38.00  || 31.40   | 26.80  || 27.40  | 28.80  || 7.00     | 5.60   || 24.70 |
>
>
>
> **Weakness** **3** **&** **Question** **2:** What are the computational costs of running your M3CoT compared to other state-of-the-art methods, as well as compared to the baseline?
>
> **Response:** We appreciate the reviewer’s thoughtful concern regarding the computational cost of M3CoT. To evaluate the tradeoff between efficiency and performance, we report inference time, token usage, and accuracy for recent CoT prompting methods, including M3CoT, evaluated on both GPT-4o and Gemini 2.0 Flash. Please note that the reported inference time may be unreliable due to variability in API response, so that it should be interpreted as a general trend rather than an exact value.
>
> As shown in the table below, compared to the baseline, M3CoT incurs an increase of approximately 45K tokens and 34.49 seconds when run on GPT-4o, and 25K tokens and 17.47 seconds on Gemini 2.0 Flash. This increase in token usage is primarily due to the refinement process that leverages three different perspectives to construct richer scene graphs. While multi-perspective reasoning requires additional computation, it enables more accurate understanding. Although the current token cost limits its use in real-time applications, we believe our multi-perspective reasoning approach provides a strong foundation for future research on efficient and scalable solutions.
>
>
> |LVLMs|Method|Token Usage|Time (s)|Acc. (%)|
> |:-|:-:|-:|-:|-:|
> |GPT-4o|Default|2,120.25|2.62|60.90|
> ||DDCoT|6,807.18|9.57|64.43|
> ||CoCoT|2,349.23|6.25|62.87|
> ||CCoT|9,225.30|22.59|63.74|
> ||**M3CoT**|**47,250.12**|**37.56**|**68.58**|
> ||||
> |Gemini 2.0|Default|790.60|2.56|59.80|
> ||DDCoT|2,752.09|5.48|60.31|
> ||CoCoT|950.71|3.84|61.09|
> ||CCoT|2,817.97|7.14|60.18|
> ||**M3CoT**|**26,164.36**|**20.03**|**66.12**|
>
>
>
> **Question** **1:** Who are the expert annotators, and what procedure did they follow to annotate the data?
>
> **Response:** The human verification stage was performed by members of our research team, which consists of the authors, all with domain expertise in vision-language reasoning and extensive experience working with ego-exo data.
> For details of the annotation procedure, including the user interface used during curation, please refer to Appendix B.3 and Figure 9 of the supplementary material. Even so, we acknowledge that additional clarification would be helpful and plan to revise the description to better convey the full curation process.
> In short, our human verification stage involved several steps to ensure quality and consistency:
>
> * We first removed questions that were ambiguous or unanswerable when both ego and exo views were provided.
>
>
> * Next, we discarded overly simple questions that could be solved through shallow pattern matching, such as cases where $A_{\text{both}}$ was correct regardless of reasoning, even if they had passed the automatic filtering stage.
>
>
> * To construct distractor options, we primarily selected responses generated under different visual input configurations. When the initial candidates were redundant or low-quality, we additionally generated options from the ego and exo views individually to supplement the distractor pool and ensure both diversity and relevance.
>
>
> * Finally, team members cross-reviewed subsets annotated by others to ensure consistency, clarity, and coverage.
>
> **Question** **3:** Why did you apply the M3CoT only to frontier models?
>
> **Response:** We appreciate the reviewer’s question on the choice of models used in our evaluation. While our main results focus on frontier models, we also conducted experiments on open-source models, including InternVL3-14B and InternVL3-8B, as shown in Appendix D.1 and Table 6 of the supplementary material.
> Each model consistently showed improved accuracy, from 53.02% to 54.79% for InternVL3-14B and from 50.44% to 53.78% for InternVL3-8B, outperforming all CoT baselines by up to 3.07%. As noted in prior works, prompting methods often struggle with smaller models due to limited reasoning capacity. Nevertheless, M3CoT consistently outperformed other approaches, demonstrating its robustness even in low-capacity settings.
>
>
>
> **Question** **4:** I think it would be great if your table of results could always report the "both" setting.
>
> **Response:** We agree with the reviewer that reporting results under the “both” setting, where questions require information from both views, provides valuable insight into a model’s ability to integrate multi-view information. At the same time, reporting results under the “any”, “ego”, and “exo” settings offers complementary information. The “any” setting assesses the model’s ability to handle redundant or overlapping information, whereas the “ego” and “exo” settings evaluate its robustness when faced with unnecessary or irrelevant views. Following this suggestion, we conducted the corresponding experiments, and the results are now presented in the table below. We are grateful for this insightful suggestion, and we will incorporate the corresponding results into the final manuscript.
>
>
> |LVLMs|Method||Any||Ego||Exo||Both||Overall|
> |-|:-:|-|-|-|-|-|-|-|-|-|-:|
> |GPT-4o|Default||65.20||63.96||58.59||39.28||56.76|
> ||DDCoT||69.71||69.37||60.83||37.68||59.40|
> ||CoCoT||66.94||64.41||61.47||43.48||59.08|
> ||CCoT||67.94||67.50||61.05||42.03||59.63|
> ||**M3CoT**||**73.45**||**74.30**||**64.23**||**52.61**||**66.15**|
> ||||
> |Gemini 2.0|Default||65.81||66.08||54.38||37.52||55.95|
> ||DDCoT||66.48||66.36||58.22||31.13||55.55|
> ||CoCoT||67.23||65.43||56.44||32.43||55.38|
> ||CCoT||65.76||66.18||55.29||43.21||57.61|
> ||**M3CoT**||**69.76**||**69.24**||**62.97**||**53.19**||**63.79**|

---

> > ### Comment · Reviewer_Gcmm · 2025-08-04
> >
> > Thank you for confirming these details. Considering that the related work is an important part of a paper, I would suggest that the author include it in the main text rather than having it in the appendix. The appendix can always have a more extensive version of the related work in case you want to cover more details regarding your approach.
> >
> > The analysis of the inference costs is really informative. I would suggest including it in your appendix and providing a clear explanation of this in your limitations section (and potentially elsewhere in your conclusions) to highlight how this boost in performance comes with an important inference cost.

---

> > > ### Author Response · Authors · 2025-08-05
> > >
> > > Thank you very much for your thoughtful feedback and for taking the time to review our work.
> > > Your suggestions have been helpful in improving the quality of our study.
> > > We are committed to thoroughly addressing your comments and implementing your recommendations faithfully.

---

### Official Review · Reviewer_itMh · 2025-07-01

**Clarity:** 3
**Significance:** 3
**Originality:** 3
**Rating:** 4
**Confidence:** 4

**Summary:**

This paper introduces a new framework for comprehensive scene understanding in large vision-language models (LVLMs) by integrating both first-person (egocentric) and third-person (exocentric) visual perspectives. The authors develop E3VQA, a benchmark consisting of 4,000 high-quality question–answer pairs grounded in synchronized ego–exo image pairs, covering tasks such as action understanding, object recognition, spatial reasoning, and numerical estimation.

**Questions:**

1. Why not use real images in Figure 1?
2. What is the difference between using the multi-view images compared with video data?
3. How to make sure the VLM can generate correct and scene graphs? Especially the presented image is quite complicated.

**Ethical Concerns:**

["NO or VERY MINOR ethics concerns only"]

**Final Justification:**

The authors have addressed my concerns by extending their benchmark to include a new dataset and introducing additional evaluation metrics to better assess scene graph quality. I appreciate these improvements, and encourage the authors to further enhance the generalization and task variety.

**Limitations:**

No.

**Paper Formatting Concerns:**

No.

**Quality:**

3

**Strengths And Weaknesses:**

# Strengths
1. The paper tackles an under-explored but highly relevant problem—comprehensive scene understanding by integrating first-person and third-person views in LVLMs. This dual-view approach goes beyond typical single-view VQA and aligns well with real-world use cases in AR/VR and robotics.
2. The E3VQA dataset is carefully constructed and covers a diverse set of tasks.

# Weaknesses
1. The E3VQA benchmark is derived from the EgoExo4D dataset, which, while large and varied, may still be limited in terms of scene diversity, activities, and environmental conditions.
2. The evaluation is limited to multiple-choice VQA with synchronized image pairs, and the number of samples is small (~4k).
3. The dataset is generated using GPT4o on images, but only the val set, no training set and finetuing existing VLMs. I think the contribution is not enough.

---

> ### Author Rebuttal · Authors · 2025-07-31
>
> We appreciate the reviewer’s invaluable comments and feedback on our paper. We tried our best to address each of your questions.
>
> **Weakness** **1:** The E3VQA benchmark is derived from the EgoExo4D dataset, which, while large and varied, may still be limited in terms of scene diversity, activities, and environmental conditions.
>
> **Response:** We understand the reviewer’s concern that relying only on the EgoExo4D dataset may not be sufficient to capture diverse real-world scenarios. In fact, to improve diversity, we extended E3VQA using LEMMA, which is a multi-view dataset focused on goal-directed daily tasks in home environments. Following the same data generation pipeline in this work, we constructed 500 new samples, evenly distributed across all categories. From experimental results on this new subset, we show that our E3VQA benchmark construction pipeline works well across different datasets, and the proposed M3CoT method provides a consistent performance gain (see table below).
>
>
>
> | LVLMs  | Method   |   |  Action  |          |        |  Object |         |        | Num.  |         |        | Spatial |         |           | Avg.  |
> |-----------|:------:|-|------------|:-------|-------|---------|:-------|-------|--------|:-------|-------|--------|:-------|---------|-------:|
> |              |          |      | Ego      | Exo    | | Ego         | Exo  |   | Ego        | Exo   |  | Ego        | Exo   |  |    |
> | GPT-4o  | Default  || 39.68     | 43.55   | | 63.49        | 72.58 |   | 56.45       | 46.03|    | 39.68       | 45.16   | | 50.83 |
> | | DDCoT || 38.10     | 48.39   | | 66.67        | 70.97   | | **64.52**       | 47.62   | | 47.62       | 38.71   | | 52.83 |
> | | CoCoT || 44.44     | 46.77   | | 69.84        | 70.97   | | 54.84       | 47.62  |  | 46.03       | **51.61**   | | 54.02 |
> | | CCoT  || 36.51     | 48.39   | | 68.25        | 69.35   | | 53.23       | 47.62  |  | 47.62       | 40.32   | | 51.41 |
> | | **M3CoT**  || **44.44**    | **51.61**  |  | **71.43**        | **75.81**  |  | 59.68       | **65.08** |    | **63.49**       | 50.00  |  | **60.19** |
> ||||
> | Gemini 2.0   | Default  ||  39.68  | 53.23   ||66.67    |74.19|   |51.61    |46.03  |  |46.03    |46.77   | |  53.03  |
> | | DDCoT  ||   39.68 |  46.77 | | 60.32   |**77.42**  | |54.84    |47.62   | |55.56    | 40.32 |  | 52.82   |
> | | CoCoT  ||  **46.03** |  48.39 | | 65.08   | 74.19 | | 50.00   |47.62  |  |47.62    |40.32    ||52.41    |
> | | CCoT   ||   44.44 |  51.61  | |66.67   |72.58  | |56.45    |**49.21**   | |46.03    |40.32  |  |53.41    |
> | | **M3CoT**   ||  44.44  |**59.68**   | |**66.67**    |72.58 |  |**56.45**    |42.86    ||**55.56**    |**50.00**  |  | **56.03**   |
>
>
>
>
>
> **Weakness** **2:** The evaluation is limited to multiple-choice VQA with synchronized image pairs, and the number of samples is small (~4k).
>
> **Response:** We appreciate the reviewer’s invaluable comment. While we understand that our multiple-choice questions can be easily converted to open-ended ones by removing answer choices, after careful thought, we decided to use the multiple-choice format to assess the model’s ability to choose the right answer from confusing alternatives. Nevertheless, we recognize the value of diverse question types (e.g., yes/no and open-ended) and will extend our benchmark.
> Regarding the concern about dataset size, we note that E3VQA is substantially larger than most existing egocentric VQA datasets. As shown in Appendix A.1 and Table 4 of the supplementary material, it is over 33 times larger than EgoVQA (120), and several times larger than EgoThink (700), EmbodiedQA (529), and OpenEQA (1.6K). While its size remains smaller than some large-scale exocentric datasets such as MSVD-QA (13K) or Social-IQ (7.5K), this gap primarily stems from the fact that third-person data is significantly more abundant than first-person data, and even more so for ego-exo paired data, making large-scale curation more challenging.
> In addition, just 3.6% of 110K GPT-4o-generated question–answer pairs were preserved in E3VQA after rigorous multi-stage filtering, and all included examples were manually verified by human annotators to ensure high quality. This strict filtering process reflects our focus on quality over quantity and ensures that the final benchmark includes challenging, well-grounded multi-view reasoning examples. We believe E3VQA provides meaningful scale and strong evaluation capacity as a benchmark for multi-view visual understanding.
>
>
> **Weakness** **3:** The dataset is generated using GPT-4o on images, but only the val set, no training set, and fine-tuning existing VLMs. I think the contribution is not enough.
>
> **Response:** We believe that the reviewer’s comment is a bit excessive, given that several other reviewers (BgYM, 2B1U, Gcmm, and 9jna) have highlighted this aspect as a strength.
>
> Please note that the primary contribution of E3VQA lies in its novel benchmark design targeting ego-exo multi-view reasoning, a setting not explored in previous VQA datasets. Please also note that our work is the first trial that systematically evaluates whether LVLMs can reason across egocentric and exocentric perspectives without task-specific fine-tuning.
> Additionally, our data generation pipeline is uniquely designed to construct plausible answer candidates by intentionally leveraging the view misreferencing behaviors of LVLMs under different visual input configurations.
> Moreover, our proposed method, M3CoT, is a training-free prompting strategy that encourages LVLMs to integrate and reason over complementary visual perspectives. It consistently improves performance by promoting cross-view alignment and understanding, without requiring any model-specific tuning. This demonstrates the potential of prompting as a plug-and-play approach to multi-view visual reasoning, particularly in scenarios where fine-tuning is not feasible.
>
>
> **Question** **1:** Why does Figure 1 not use real images?
>
> **Response:** In Figure 1, we intentionally used a conceptual illustration rather than real images to provide an intuitive explanation of our framework. We believe this synthetic image effectively highlights the complementary nature of the ego and exo views by illustrating how their fields of view differ, something that is difficult to depict clearly with real images. To avoid confusion, we will clarify in the caption that Figure 1 is a conceptual illustration and is not part of the actual dataset. For real examples from the dataset, we will kindly refer the reader to subsequent figures throughout the paper.
>
>
> **Question** **2:** What is the difference between using the multi-view images compared with video data?
>
> **Response:** Although multi-view images and video data seem to be similar since they consist of multiple images, they differ in how the visual information is captured and ordered. Video captures a continuous sequence of frames from a single viewpoint over time, preserving temporal order. In contrast, multi-view images are captured simultaneously from different angles at a single time point, where the sequence order is not meaningful. While video can offer complementary views over time as the camera moves, potentially covering different angles of the scene, the context may change between frames, making it difficult to align information across time. As a result, video cannot fully replace multi-view images, which capture different viewpoints at the same moment without temporal drift.
>
>
> **Question** **3:**  How to make sure the VLM can generate correct scene graphs? The presented image is quite complicated.
>
> **Response:** We appreciate the reviewer’s insightful comment. While related analysis is included in Appendix D.2 of the supplementary material, we provide an extended explanation here to enhance clarity and better highlight its significance.
> Scene graph quality is often assessed by computing the intersection-over-union (IoU) with ground-truth scene graphs. However, overly exhaustive graphs that capture all visual details may not always be beneficial for answering questions accurately. To better assess the practical utility of scene graphs generated by M3CoT, we introduce two complementary metrics:
> * False discovery rate (FDR) is measured as the proportion of incorrect elements among the model's predicted scene graph components, such as object classes, relations, and attributes. This reflects how accurately the model captures the visual content.
> * Answer accuracy is measured based on the VQA performance when the generated scene graph is used as a prompt. This reflects how well the scene graph supports downstream reasoning.
>
> To evaluate the relationship between FDR and answer accuracy of the generated scene graph, we constructed a subset of 120 samples from E3VQA, equally distributed across all question categories. For this subset, we evaluated the scene graph before and after the refinement stage of M3CoT. The FDR of the scene graph decreased from 9.37% to 6.21%, with an increase in answer accuracy from 46.88% to 60.42%. This result demonstrates that our refinement stage not only corrects errors in the scene graph but also organizes the information in a way that better supports question answering.
>
> We will add these supplements to the revised manuscript. We hope our responses can address your concerns.

---

> > ### Author Response · Authors · 2025-08-07
> >
> > Thank you once again for taking the time to review our paper and for providing valuable comments.
> > Following your feedback, we have sincerely worked to address your concerns.
> >
> > As the discussion deadline approaches, please don’t hesitate to let us know if you have any further questions. We would be glad to respond and are committed to revising the paper further to reflect your suggestions and enhance its quality. Thank you for your time and thoughtful consideration.

---

> > ### Comment · Reviewer_itMh · 2025-08-07
> >
> > The authors have addressed my concerns by extending their benchmark to include a new dataset and introducing additional evaluation metrics to better assess scene graph quality. I appreciate these improvements, and encourage the authors to further enhance the generalization and task variety.

---

> > > ### Author Response · Authors · 2025-08-08
> > >
> > > We greatly appreciate your thoughtful feedback and the time you took to review our work. Your suggestions have been instrumental in clarifying the contributions of our study.
> > >
> > > We are dedicated to carefully addressing your comments and diligently incorporating your recommendations into the revised manuscript.

---

### Official Review · Reviewer_2B1U · 2025-07-03

**Clarity:** 3
**Significance:** 3
**Originality:** 3
**Rating:** 5
**Confidence:** 4

**Summary:**

his paper tackles the challenge of enabling large vision-language models (LVLMs) to jointly reason over first-person (egocentric) and third-person (exocentric) views. The authors introduce E3VQA, a new benchmark comprising 4K high-quality question–answer pairs grounded in synchronized ego–exo image pairs, covering tasks such as action understanding, attribute recognition, spatial reasoning, and counting. They also propose M3CoT, a training-free prompting method that constructs multi-perspective scene graphs (Ego&Exo, Ego→Exo, Exo→Ego), which are progressively unified to form a complete scene representation. Experiments on leading closed-source (GPT-4o, Gemini) and open-source LVLMs show that M3CoT consistently outperforms strong chain-of-thought baselines (e.g., CCoT) with up to 5.9% absolute accuracy improvement, especially in numerical reasoning tasks.

**Questions:**

Since M3CoT applies a full multi-perspective prompting pipeline to all questions, would it be possible to selectively activate only certain views or strategies based on question type or view relevance (e.g., skip full graph fusion for single-view solvable questions) to reduce inference overhead?

**Ethical Concerns:**

["NO or VERY MINOR ethics concerns only"]

**Final Justification:**

I appreciate the response from the author. I would keep my positive rating.

**Limitations:**

yes

**Quality:**

3

**Strengths And Weaknesses:**

Strengths:
1 the paper the well-writen and easy to follow.
2 E3VQA fills an evaluation gap in video understanding by targeting egocentric–exocentric joint understanding with carefully curated multiple-choice QA pairs. Its design rigorously tests multi-view integration rather than single-view shortcuts.
3: M3CoT enhances reasoning across perspectives without additional model training, and achieves consistent performance gains over state-of-the-art CoT prompting baselines, especially in spatial and numerical questions.

Weaknesses
The approach relies on synchronized ego–exo image pairs, limiting its applicability to real-world settings where such pairing may be noisy, missing, or unavailable.
M3CoT involves multiple scene graph generations and iterative refinement. The cost and stability of this prompting procedure across different LVLMs are not fully analyzed, and it may pose scalability concerns.

---

> ### Author Rebuttal · Authors · 2025-07-31
>
> We appreciate the reviewer’s invaluable comments and feedback on our paper. We tried our best to address each of your questions.
>
> **Weakness** **1:** Dependence on synchronized ego-exo pairs could limit real-world use where such data is incomplete or noisy.
>
> **Response:** Thank you for the reviewer’s insightful question. While we acknowledge that maintaining synchronization between ego and exo views can be challenging in real-world settings, our analysis suggests that minor temporal mismatches (e.g., a few seconds) have minimal impact on the correctness of the ground truth. To examine the effect of temporal mismatches on the ground truth in E3VQA, we randomly sample 80 examples (10 from each category) and analyze how the ground truth changes when the ego and exo views are misaligned by 1 to 3 seconds. As shown in the table below, when the synchronization gap is 1 second, 99% of the ground truth answers remained unchanged. Even with a 3-second gap, 94% of the ground truth remained consistent. In addition, regarding the question of potential noise or missing information in one of the views, we expect that the use of complementary multi-view inputs can itself mitigate such issues. Even if part of the information is missing or corrupted in one view, the model can leverage the other view to fill in the gaps. Therefore, we expect that our framework can provide more accurate and robust answers than single-view baselines, even when there are small synchronization issues or partial data loss.
>
>
>
> **Weakness** **2:** M3CoT's multi-step prompting lacks analysis of cost, stability, and scalability across LVLMs.
>
> **Response:** We appreciate the reviewer’s thoughtful concern regarding the computational cost of M3CoT. To evaluate the tradeoff between efficiency and performance, we report inference time, token usage, and accuracy for recent CoT prompting methods, including M3CoT, evaluated on both GPT-4o and Gemini 2.0 Flash. Please note that the reported inference time may be unreliable due to variability in API response, so that it should be interpreted as a general trend rather than an exact value.
>
> As shown in the table below, compared to the baseline, M3CoT incurs an increase of approximately 45K tokens and 34.49 seconds when run on GPT-4o, and 25K tokens and 17.47 seconds on Gemini 2.0 Flash. This increase in token usage is primarily due to the refinement process that leverages three different perspectives to construct richer scene graphs. While multi-perspective reasoning requires additional computation, it enables more accurate understanding. Although the current token cost limits its use in real-time applications, we believe our multi-perspective reasoning approach provides a strong foundation for future research on efficient and scalable solutions.
>
>
> |LVLMs|Method|Token Usage|Time (s)|Acc. (%)|
> |:-|:-:|-:|-:|-:|
> |GPT-4o|Default|2,120.25|2.62|60.90|
> ||DDCoT|6,807.18|9.57|64.43|
> ||CoCoT|2,349.23|6.25|62.87|
> ||CCoT|9,225.30|22.59|63.74|
> ||**M3CoT**|**47,250.12**|**37.56**|**68.58**|
> ||||
> |Gemini 2.0|Default|790.60|2.56|59.80|
> ||DDCoT|2,752.09|5.48|60.31|
> ||CoCoT|950.71|3.84|61.09|
> ||CCoT|2,817.97|7.14|60.18|
> ||**M3CoT**|**26,164.36**|**20.03**|**66.12**|
>
>
>
>
> **Question** **1:** Can M3CoT optimize inference by dynamically selecting views or reasoning paths based on question demands?
>
> **Response:** Thank you for the insightful question. We truly agree that integrating a query-driven view selection strategy into M3CoT could make inference much more efficient, especially in cases where the relevant view can be inferred from the question. That said, we would like to mention that we intentionally design E3VQA to avoid such cases so that the model is encouraged to consider both views when answering. Therefore, selecting the appropriate view based solely on the query may be challenging in many E3VQA examples. Nonetheless, we fully support the idea that your proposed strategy would be highly effective in practical applications beyond the benchmark setting, where efficiency is critical and many real-world questions naturally provide cues that make view selection more predictable.
>
> We will add these supplements to the revised manuscript. We hope our responses can address your concerns.

---

> > ### Author Response · Authors · 2025-08-07
> >
> > Thank you once again for taking the time to review our paper and for providing valuable comments.
> > Following your feedback, we have sincerely worked to address your concerns.
> >
> > As the discussion deadline approaches, please don’t hesitate to let us know if you have any further questions. We would be glad to respond and are committed to revising the paper further to reflect your suggestions and enhance its quality. Thank you for your time and thoughtful consideration.

---

### Official Review · Reviewer_BgYM · 2025-07-03

**Clarity:** 3
**Significance:** 3
**Originality:** 3
**Rating:** 4
**Confidence:** 4

**Summary:**

This paper introduces E3VQA, a novel benchmark designed to evaluate large vision-language models (LVLMs) on multi-view question answering that combines egocentric and exocentric perspectives. Recognizing the complementary nature of both views, fine-grained hand-object interactions from ego and spatial context from exo. The authors argue that current LVLMs fail to reason effectively across both.

They additionally propose M3CoT, a training-free prompting technique that uses multi-perspective scene graphs and iteratively fuses them for improved reasoning. Extensive experiments on E3VQA show that M3CoT yields consistent and significant performance gains over recent prompting baselines across multiple LVLMs including GPT-4o and Gemini 2.0 Flash.

**Questions:**

1. Can M3CoT be applied in real-time systems (e.g., AR devices), or is its inference-time cost too high? Consider discussing latency and compute implications of multi-agent prompting.
2. Is there a risk of conflicting information between scene graphs (e.g., ego says “holding a cup,” exo says “not holding anything”)? How is this resolved during refinement?
3. Do you leverage any visual feature alignment or object detection models (e.g., pose estimation) to help match entities across views, or is everything handled purely via language prompting?
4. Do you expect E3VQA-trained models or the M3CoT prompting approach to generalize to other ego-exo datasets like Charades-Ego or OpenEQA?
5. Some qualitative examples of M3CoT failure or ambiguous scene interpretations would help readers understand the boundary of your method's capability.
6. How does your model handle ambiguous or inherently subjective questions (e.g., “What am I likely to do next?”) where multiple answers might be plausible?

**Ethical Concerns:**

["NO or VERY MINOR ethics concerns only"]

**Limitations:**

Yes

**Quality:**

3

**Strengths And Weaknesses:**

### Strengths:

1. Timely and Novel Benchmark: It is a first-of-its-kind dataset focusing on ego-exo reasoning, addressing a growing need in augmented reality, embodied AI, and multimodal understanding.
2. The dataset includes 4K QA pairs evenly distributed across four categories (pose, object, numerical, spatial), with fine-grained filtering and expert human verification.
3. The multi-agent, multi-perspective scene graph construction and refinement mechanism is original, cleverly designed, and provides demonstrable performance improvements.
4. M3CoT significantly outperforms multiple prior CoT prompting methods across a range of LVLMs and question types. The performance analysis is thorough and includes ablations.
5. The paper includes implementation details, error bars, dataset access, compute info, and a discussion of limitations and ethical concerns.

### Weaknesses:
1. While the E3VQA benchmark is impactful, its reliance on a single source dataset (EgoExo4D) may limit its diversity and generalization across real-world applications.
2. Although M3CoT is "training-free," its reliance on multiple inference passes with different prompt agents may incur significant inference cost, which isn’t discussed in depth.
3. While the benchmark generation pipeline is verified, there’s no qualitative user study or human preference test to assess the final outputs or user experience improvements from M3CoT.
4. While many open-source models are evaluated, M3CoT’s effectiveness is demonstrated only on two closed-source models. It would strengthen the paper to evaluate M3CoT on open-source LLMs (e.g., InternVL3).

---

> ### Author Rebuttal · Authors · 2025-07-31
>
> We appreciate the reviewer’s invaluable comments and feedback on our paper. We tried our best to address each of your questions.
>
> **Weakness** **1:**  E3VQA's reliance on a single dataset, EgoExo4D, may hinder its diversity and real-world applications.
>
>
> **Response:** We understand the reviewer’s concern that relying only on the EgoExo4D dataset may not be sufficient to capture diverse real-world scenarios. In fact, to improve diversity, we extended E3VQA using LEMMA, which is a multi-view dataset focused on goal-directed daily tasks in home environments. Following the same data generation pipeline in this work, we constructed 500 new samples, evenly distributed across all categories. From experimental results on this new subset, we show that our E3VQA benchmark construction pipeline works well across different datasets, and the proposed M3CoT method provides a consistent performance gain (see table below).
>
>
>
> |LVLMs|Method||Action|||Object|||Num.|||Spatial|||Avg.|
> |-|:-:|-|-|:-|-|-|:-|-|-|:-|-|-|:-|-|-:|
> ||||Ego|Exo||Ego|Exo||Ego|Exo||Ego|Exo|||
> |GPT-4o|Default||39.68|43.55||63.49|72.58||56.45|46.03||39.68|45.16||50.83|
> ||DDCoT||38.10|48.39||66.67|70.97||**64.52**|47.62||47.62|38.71||52.83|
> ||CoCoT||44.44|46.77||69.84|70.97||54.84|47.62||46.03|**51.61**||54.02|
> ||CCoT||36.51|48.39||68.25|69.35||53.23|47.62||47.62|40.32||51.41|
> ||**M3CoT**||**44.44**|**51.61**||**71.43**|**75.81**||59.68|**65.08**||**63.49**|50.00||**60.19**|
> ||||
> |Gemini 2.0|Default||39.68|53.23||66.67|74.19||51.61|46.03||46.03|46.77||53.03|
> ||DDCoT||39.68|46.77||60.32|**77.42**||54.84|47.62||55.56|40.32||52.82|
> ||CoCoT||**46.03**|48.39||65.08|74.19||50.00|47.62||47.62|40.32||52.41|
> ||CCoT||44.44|51.61||66.67|72.58||56.45|**49.21**||46.03|40.32||53.41|
> ||**M3CoT**||44.44|**59.68**||**66.67**|72.58||**56.45**|42.86||**55.56**|**50.00**||**56.03**|
>
>
>
>
>
>
> **Weakness** **2** **&** **Question** **1:**  M3CoT’s reliance on multiple inference passes with different prompt agents may incur significant inference cost.
>
> **Response:** We appreciate the reviewer’s thoughtful concern regarding the computational cost of M3CoT. To evaluate the tradeoff between efficiency and performance, we report inference time, token usage, and accuracy for recent CoT prompting methods, including M3CoT, evaluated on both GPT-4o and Gemini 2.0 Flash. Please note that the reported inference time may be unreliable due to variability in API response, so that it should be interpreted as a general trend rather than an exact value.
>
> As shown in the table below, compared to the baseline, M3CoT incurs an increase of approximately 45K tokens and 34.49 seconds when run on GPT-4o, and 25K tokens and 17.47 seconds on Gemini 2.0 Flash. This increase in token usage is primarily due to the refinement process that leverages three different perspectives to construct richer scene graphs. While multi-perspective reasoning requires additional computation, it enables more accurate understanding. Although the current token cost limits its use in real-time applications, we believe our multi-perspective reasoning approach provides a strong foundation for future research on efficient and scalable solutions.
>
>
> |LVLMs|Method|Token Usage|Time (s)|Acc. (%)|
> |:-|:-:|-:|-:|-:|
> |GPT-4o|Default|2,120.25|2.62|60.90|
> ||DDCoT|6,807.18|9.57|64.43|
> ||CoCoT|2,349.23|6.25|62.87|
> ||CCoT|9,225.30|22.59|63.74|
> ||**M3CoT**|**47,250.12**|**37.56**|**68.58**|
> ||||
> |Gemini 2.0|Default|790.60|2.56|59.80|
> ||DDCoT|2,752.09|5.48|60.31|
> ||CoCoT|950.71|3.84|61.09|
> ||CCoT|2,817.97|7.14|60.18|
> ||**M3CoT**|**26,164.36**|**20.03**|**66.12**|
>
>
>
>
> **Weakness** **3:** M3CoT lacks user studies or human evaluations to validate its output quality or user preference.
>
> **Response:** In our experiments, M3CoT generates an answer from the choices (e.g., A, B, C, or D), as we design E3VQA in a multiple-choice format to simplify the evaluation process. Nonetheless, extending E3VQA to support open-ended responses and evaluating M3CoT's outputs through a user study or human preference assessment is indeed a valuable future direction. We sincerely appreciate this suggestion and consider it a meaningful avenue for future research.
>
> **Weakness** **4:** M3CoT is only tested on closed-source models; evaluating it on open-source LLMs like InternVL3 would strengthen the work.
>
> **Response:** In Appendix D.1 and Table 6, we compare our proposed M3CoT prompting technique with existing CoT methods on open-source LVLMs, specifically InternVL3-8B and InternVL3-14B. When the CoT-based approach is applied to small-sized models, it is known to be less effective due to their limited reasoning capabilities. However, we observe that M3CoT performs well even with small-sized models, which is not the case for conventional CoT-based approaches.
>
>
>
>
> |LVLMs|Method||Action|||Object|||Num.|||Spatial|||Avg.|
> |-|:-:|-|-|:-|-|-|:-|-|-|:-|-|-|:-|-|-:|
> ||||Ego|Exo||Ego|Exo||Ego|Exo||Ego|Exo|||
> |InternVL3-14B|Default||44.73|54.93||68.13|73.73||35.60|**53.00**||45.67|48.33||53.02|
> ||CCoT||44.60|58.40||65.27|73.80||**37.80**|50.00||46.27|48.80||53.12|
> ||**M3CoT**||**45.87**|**60.00**||**70.60**|**75.73**||35.07|50.87||**50.80**|**49.40**||**54.79**|
> ||||
> |InternVL3-8B|Default||43.70|54.90||64.80|70.30||35.90|45.20||42.10|46.60||50.44|
> ||CCoT||44.00|55.00||63.60|68.30||35.50|**51.20**||43.40|44.70||50.71|
> ||**M3CoT**||**45.50**|**57.20**||**68.20**|**71.20**||**37.60**|47.50||**53.80**|**49.20**||**53.78**|
>
>
>
>
> **Question** **2:** Is there a risk caused by conflicting scene graphs between different perspectives, and how is it resolved during refinement?
>
> **Response:** You're absolutely right to highlight this potential risk, and we fully acknowledge its importance. In fact, addressing this issue is precisely the motivation behind our refinement stage. The primary goal of the refinement stage is to resolve inconsistencies between scene graphs generated from different perspectives. When the Ego2Exo and Exo2Ego perspectives produce conflicting scene graphs, the refinement stage encourages each perspective to “reconsider” its own scene graph by referencing the other’s scene graph and the corresponding ego-exo image pair. This process can be interpreted as encouraging the model to overcome its own bias. Details of the prompt used during this refinement stage can be found in Figure 37. We intend to include additional qualitative examples of such conflict resolution in the final manuscript.
> In addition, the effectiveness of the refinement stage is indirectly demonstrated by the results shown in Appendix D.2 and Figure 9. Compared to iteration 0, iteration 1 (e.g., after the refinement stage) exhibits notable improvements in accuracy, both in the individual predictions from each perspective and in the final answer obtained via majority voting. These results support the effectiveness of the refinement stage in resolving conflicts between perspectives.
>
> **Question** **3:** Do you use visual alignment tools (e.g., pose estimation), or rely solely on language prompting?
>
> **Response:** We do not use any external guidance. Our primary goal is to explore how far the LVLM’s inherent capabilities can be utilized through prompting alone, without relying on external modules or tools. We expect that M3CoT might achieve further accuracy gains with the addition of external information.
>
> **Question** **4:** Can M3CoT generalize to other ego-exo datasets like Charades-Ego or OpenEQA?
>
> **Response:** Thank you for the insightful question. We expect that models trained on E3VQA or using the proposed M3CoT prompting approach can be applied to other datasets such as Charades-Ego and OpenEQA. While Charades-Ego involves asynchronous ego-exo pairs and OpenEQA focuses on embodied question answering, both datasets fundamentally require the model to extract and integrate information from multiple images to answer the questions. Given this shared requirement, we expect that our work would be effective and applicable in such datasets.
>
>
> **Question** **5:** Showing failure cases would help readers better understand M3CoT’s capabilities.
>
> **Response:** Thank you for the valuable feedback. Unfortunately, please note that the addition of figures is not allowed this year. We will incorporate qualitative examples analyzing M3CoT's failure cases into the revised manuscript. As you suggested, we believe these examples will help readers better understand the limitations and strengths of our proposed method.
>
> **Question** **6:** How does your model handle subjective questions with multiple plausible answers?
>
>
> **Response:** Regarding the E3VQA benchmark, we try our best to avoid ambiguous or subjective questions and answers. With respect to M3CoT, we design it to generate accurate scene graphs that not only capture multi-view information but are also relevant to the given question. These scene graphs help the model generate image-grounded answers, even for ambiguous queries like 'What am I likely to do next?'". For instance, consider a scenario where both “slicing a tomato” and “washing a cucumber” are plausible next actions. If the scene graph generated by M3CoT includes elements such as <I, holding, tomato>, and  <cucumber, on, table>, the model is more likely to answer “slicing a tomato”, as this action is more directly supported by the visual evidence in the scene. In summary, M3CoT guides the model to focus on visual cues rather than prior knowledge, enabling more image-grounded responses even in ambiguous situations.
>
>
>
> We will add these supplements to the revised manuscript. We hope our responses can address your concerns.

---

> > ### Comment · Reviewer_BgYM · 2025-08-05
> >
> > Thank you for the detailed and well-organized rebuttal.
> >
> > The inclusion of LEMMA-based E3VQA samples strengthens the generalization argument, and the added results on InternVL3 are appreciated. I also found your clarification on refinement mechanisms for conflicting scene graphs and your quantitative breakdown of token costs helpful.
> >
> > The lack of user studies and qualitative failure analysis remains a gap, but your intention to include these in future versions is noted. Overall, the rebuttal addresses my key concerns, and I continue to view this as a good contribution. I stand by my original evaluation.

---

> > > ### Author Response · Authors · 2025-08-07
> > >
> > > Thank you sincerely for your insightful feedback and for dedicating your time to review our work. Your suggestions have been invaluable in enhancing the quality of our study. We are fully committed to carefully addressing your comments and incorporating your recommendations with diligence.

---

### Note · Authors · 2025-08-14

We sincerely thank all reviewers for their useful comments and thoughtful suggestions.
In the following, we summarize the key points in the discussion phase.

---

### **Strengths Agreed by the Reviewers**

Reviewers thankfully recognized several strengths of our work, including timely and novel benchmark for ego-exo reasoning (Reviewers BgYM, 2B1U, and itMh), rigorous and well-verified dataset construction pipeline (Reviewers BgYM, itMh, Gcmm, and 9jna), original and effective prompting technique achieving consistent performance gains (Reviewers BgYM, 2B1U, Gcmm, and 9jna), thorough and well-designed experimental evaluation across diverse models and tasks (Reviewers BgYM and Gcmm), and also the clear presentation (Reviewers 2B1U, Gcmm, and 9jna).


### **Reviewers’ Concerns and Our Responses**

**1. Diversity and Generalization of the E3VQA Benchmark**

To enhance diversity and generalization quality, we extended E3VQA with the LEMMA dataset. Newly obtained results on this subset match well with our main findings: 1) recent LVLMs struggle with multi-view reasoning, and 2) M3CoT consistently outperforms other prompting techniques.

**2. Inference Cost Analysis of M3CoT**

We compared token usage and inference time between M3CoT and recent prompting techniques. Although the cost and inference time increase slightly, M3CoT substantially improves multi-view reasoning performance without requiring extra training.

**3. Evaluation of Scene Graph Quality**

We evaluated scene graph quality before and after the refinement, from which we observed that scene graph errors decrease, leading to an overall performance improvement during the refinement stage.

**4. Evaluation of M3CoT on Open-Source Models**

We reported the performance of M3CoT on open-source models of varying sizes. The results demonstrate that it consistently delivers performance gains across models of different scales.

**5. Impact of Temporal Misalignment on the Multi-View Framework**

We analyzed our framework’s robustness by examining changes in the ground truth when ego and exo views are temporally misaligned. Small mismatches (~3s) cause only minimal changes, demonstrating its stability against minor synchronization errors.

---

We truly appreciate the reviewers’ and ACs’ time and effort, and we will reflect these points and the additional issues addressed in the rebuttal period in the final manuscript.

---

### Decision · Program_Chairs · 2025-09-17

**Decision:**

Accept (spotlight)

**Comment:**

**1. Summary of Scientific Claims and Findings**

This paper introduces E3VQA, a novel benchmark designed to evaluate large vision-language models (LVLMs) on multi-view question answering (QA), with a major argument that integrating egocentric (first-person) and exocentric (third-person) perspectives  can provide complementary information such as global scene layout and object visibility to LVLMs. The authors found that current LVLMs struggle to reason effectively across these complementary views (e.g., fine-grained hand-object interactions from ego views and spatial context from exo views). Additionally, they proposed M3CoT, a training-free prompting technique that uses multi-perspective scene graphs (ego-exo, ego→exo, exo→ego) and iteratively fuses them to enhance cross-view reasoning. Experiments on E3VQA (4K expert-verified QA pairs across four categories: pose, object, numerical, spatial) showed that M3CoT consistently outperforms existing prompting baselines across multiple LVLMs, including GPT-4o and Gemini 2.0 Flash.

**2. Major Strengths**

2.1 Novel and rigorous benchmark. E3VQA fills a critical gap in evaluating multi-view (ego-exo) reasoning, with 4K high-quality, expert-verified QA pairs that ensure reliance on cross-view integration rather than single-view shortcuts.

2.2 Effective training-free technique. M3CoT yields significant performance gains over state-of-the-art prompting baselines across LVLMs, particularly in spatial and numerical reasoning tasks.

2.3 Thorough evaluation. The paper includes extensive experiments on both closed-source (e.g., GPT-4o) and open-source (e.g., InternVL3) LVLMs, with ablations, error analysis, and transparent details (implementation, dataset access, compute).

2.4 Clarity and rigor. The paper is well-written, with a clear narrative, rigorous data collection pipeline, and discussion of limitations/ethical concerns.

**3. Major Weaknesses**

3.1 Benchmark limitations. E3VQA relies solely on EgoExo4D, limiting diversity and generalization. Its size (~4K) and focus on multiple-choice QA may restrict broader applicability.

3,2 M3CoT scalability. While training-free, M3CoT’s multiple inference passes raise concerns about high inference cost and scalability, which are not deeply analyzed. Its effectiveness is under-tested on open-source LVLMs.

3.3 Missing analyses. No qualitative user studies, human preference tests, or text-only baselines to assess output quality or data biases. Insufficient discussion of ego-exo integration challenges compared to other multi-view scenarios, and limited analysis of VLM failure patterns.

3.4 Structural gaps: Lack of a dedicated related work section to contextualize E3VQA against existing egocentric QA datasets.

**4. The most important reasons to accept**

This paper received ratings of (4, 5, 4, 5, 5), whose avage rating is the highest among my batch of 14 submissions. The paper has good novelty and the developed benchmark is expected to have a high impact to the field of vision-language models.

**5. Summary of discussions**

There has been some discussions between the authors and reviewers. Most of the questions from the reviewers have been addressed by the authors. After the rebutal, Reviewer 9jna has raised his rating from 4 to 5.